# Investigating the Ability of PINNs To Solve Burgers' PDE Near Finite-Time BlowUp

## Abstract

Physics Informed Neural Networks (PINNs) have been achieving ever newer feats of solving complicated PDEs numerically while offering an attractive trade-off between accuracy and speed of inference. A particularly challenging aspect of PDEs is that there exist simple PDEs which can evolve into singular solutions in finite time starting from smooth initial conditions. In recent times some striking experiments have suggested that PINNs might be good at even detecting such finite-time blow-ups. In this work, we embark on a program to investigate this stability of PINNs from a rigorous theoretical viewpoint. Firstly, we derive generalization bounds for PINNs for Burgers' PDE, in arbitrary dimensions, under conditions that allow for a finite-time blow-up. Then we demonstrate via experiments that our bounds are significantly correlated to the $\ell_2$-distance of the neurally found surrogate from the true blow-up solution, when computed on sequences of PDEs that are getting increasingly close to a blow-up.

## 1 Introduction

Partial Differential Equations (PDEs) are used for modeling a large variety of physical processes from fluid dynamics to bacterial growth to quantum behaviour at the atomic scale. But differential equations that can be solved in "closed form," that is, by means of a formula for the unknown function, are the exception rather than the rule. Hence over the course of history, many techniques for solving PDEs have been developed. However, even the biggest of industries still find it extremely expensive to implement the numerical PDE solvers – like airplane industries aiming to understand how wind turbulence pattern changes with changing aerofoil shapes, (Jameson et al., 2002) need to choose very fine discretizations which can often increase the run-times prohibitively.

In the recent past, deep learning has emerged as a competitive way to solve PDEs numerically. We note that the idea of using nets to solve PDEs dates back many decades Lagaris et al. (1998) (Broomhead & Lowe, 1988). In recent times this idea has gained significant momentum and "AI for Science" (Karniadakis et al., 2021) has emerged as a distinctive direction of research. Some of the methods at play for solving PDEs neurally (E et al., 2021) are the Physics Informed Neural Networks (PINNs) paradigm (Raissi et al., 2019) (Lawal et al., 2022), "Deep Ritz Method" (DRM, Yu et al. (2018)), "Deep Galerkin Method" (DGM, Sirignano & Spiliopoulos (2018)) and many further variations that have been developed of these ideas, (Kaiser et al., 2021; Erichson et al., 2019; Wandel et al., 2021; Li et al., 2022; Salvi et al., 2022). An overarching principle that many of these implement is to try to constrain the loss function by using the residual of the PDE to be solved.

These different data-driven methods of solving the PDEs can broadly be classified into two kinds, **(1)** ones which train a single neural net to solve a specific PDE and **(2)** operator methods – which train multiple nets in tandem to be able to solve a family of PDEs in one shot. (Fan et al., 2019; Lu et al., 2021; 2022; Wang et al., 2021b) The operator methods are particularly interesting when the underlying physics is not known and the state-of-the-art approaches of this type can be seen in works like Raonić et al. (2023), Kovachki et al. (2023) and Fan et al. (2019).

For this work, we focus on the PINN formalism from Raissi et al. (2019). Many studies have demonstrated the success of this setup in simulating complex dynamical systems like the Navier-Stokes PDE (Arthurs & King, 2021; Wang et al., 2020; Eivazi et al., 2022), the Euler PDE (Wang

---

Implementation is available at https://anonymized

et al., 2022d), descriptions of shallow water wave by the Korteweg-De Vries PDEs (Hu et al., 2022a) and many more.

Work in Mishra & Molinaro (2022); De Ryck et al. (2022) has provided the first-of-its-kind bounds on the generalization error of PINNs for approximating various standard PDEs, including the Navier-Stokes' PDE. Such bounds strongly motivate why minimization of the PDE residual at collocation points can be a meaningful way to solve the corresponding PDEs. However, the findings and analysis in Krishnapriyan et al. (2021); Wang et al. (2021a) point out that the training dynamics of PINNs can be unstable and failure cases can be found among even simple PDE setups. It has also been studied that when trivial solutions can exist for the PDE, the PINN training can get stuck at those solutions Rohrhofer et al. (2022); Cheng Wong et al. (2022). Work in Wang et al. (2022b) has shown that traditional ways of solving PINNs can violate causality.

However, in all the test cases above the target solutions have always been nice functions. But an interesting possibility with various differential equations representing the time dynamics of some system is that their solution might have a finite-time blow-up. Blow-up is a phenomena where the solution becomes infinite at some points as t approaches a certain time $T < \infty$, while the solution is well-defined for all $0 < t < T$ i.e.

$$\sup_{x \in D} |\boldsymbol{u}(\boldsymbol{x}, t)| \to \infty \quad as \quad t \to T^-$$

One can see simple examples of this fascinating phenomenon, for example, for the following ODE $\frac{du}{dt} = u^2$, $u(0) = u_0$, $u_0 > 0$ it's easy to see that it's solution blows-up at $t = \frac{1}{u_0}$. Wintner's theorem (Wintner, 1945) provided a sufficient condition for a very generic class of ODEs for the existence of a well-defined solution for them over the entire time-domain, in other words, the non-existence of a finite-time blowup. More sophisticated versions of such sufficient conditions for global ODE solutions were subsequently developed Cooke (1955) and Pazy (1983) (Theorem 3.3). Non-existence of finite-time blow-ups have also been studied in control theory (Lin et al., 1996) under the name of "forward completeness" of a system.

The existence of a blow-up makes solving PDEs difficult to solve for classical approximation methods. There is a long-standing quest in numerical methods of PDE solving to be able to determine the occurrence, location and nature of finite time blow-ups (Stuart & Floater, 1990). A much investigated case of blow-up in PDE is for the exponential reaction model $\boldsymbol{u}_t = \Delta \boldsymbol{u} + \lambda e^{\boldsymbol{u}}$, $\lambda > 0$ which was motivated as a model of combustion under the name Frank-Kamenetsky equation. The nature of blow-up here depends on the choice of $\lambda$, the initial data and the domain. Another such classical example is $\boldsymbol{u}_t = \Delta \boldsymbol{u} + \boldsymbol{u}^p$ and both these semi-linear equations were studied in the seminal works Fujita (1966; 1969) which pioneered systematic research into finite-time blow-ups of PDEs.

To the best of our knowledge, the behaviour PINNs in the proximity of finite-time blow-up has not received adequate attention in prior work on PINNs. We note that there are multiple real-world phenomena whose PDE models have finite-time blow-ups and these singularities are known to correspond to practically relevant processes – such as in chemotaxis models (Herrero & Velazquez, 1997; He & Tadmor, 2019; Chen et al., 2022; Tanaka, 2023) and thermal-runoff models (Bebernes & Kassoy, 1981; Lacey, 1983; Dold, 1991; Herrero & Velázquez, 1993; Lacey, 1995).

In light of the recent rise of methods for PDE solving by neural nets, it raises a curiosity whether the new methods, in particular PINNs, can be used to reliably solve PDEs near such blow-ups. While a general answer to this is outside the scope of this work, *we derive theoretical risk bounds for PINNs which are amenable to be tested against certain analytically describable finite-time blow-ups. Additionally, we give experiments to demonstrate that our bounds retain non-trivial insight even when tested in the proximity of such singularities.*

In Wang et al. (2022d), thought provoking experimental evidence was given that PINNs could potentially discover PDE solutions with blow-up even when their explicit descriptions are not known. Hence inspired, here we embark on a program to understand this interface from a rigorous viewpoint and show how well the theoretical risk bounds correlate to their experimentally observed values - in certain blow-up situations. As our focus point, we will use reduced models of fluid dynamics, i.e Burgers' PDE in one and two spatial dimensions. The choice of our test case is motivated by the fact that these PDE setups have analytic solutions with blow-up – as is necessary to do a controlled study of PINNs facing such a situation. We note that it is otherwise very rare to know exact fluid-like solutions which blow-up in finite-time (Tao, 2016a;b)

## 1.1 Informal Summary of Our Results

At the very outset, we note that to the best of our knowledge there are no available off-the-shelf generalization bounds for any setup of PDE solving by neural nets where the assumptions being made include any known analytic solution with blow-up for the corresponding PDE. So, as a primary step we derive new risk bounds for Burgers's PDE in Theorem 3.1 and Theorem 3.2, where viscosity is set to zero and its boundary conditions are consistent with finite-time blow-up cases of Burgers' PDE that we eventually want to test on. We note that despite being designed to cater to blow-up situations, the bound in Theorem 3.2 is also "stable" in the sense of Wang et al. (2022a).

Our experiments reveal that for our test case with Burgers' PDE while the theoretical error bounds we derive are vacuous (as is routine for neural net generalization bounds), somewhat surprisingly they do maintain a non-trivial amount of correlation with the $L^2$-distance of the derived solution from the true solution. The plot in Figures 1 and 5 vividly exhibit the presence of this strong correlation between the derived bounds and the true risk despite the experiments being progressively made on time domains such that the true solution is getting arbitrarily close to becoming singular.

A key feature of our approach to this investigation is that we do not tailor our theory to the experimental setups we test on later. We posit that this is a fair way to evaluate the reach of PINN theory whereby the theory is built such that it caters to any neural net and any solution of the PDE while these generically derived bounds get tested on the hard instances. [2]

## 1.2 A Review of the Framework of Physics-Informed Neural Networks

Consider the following specification of a PDE satisfied by an appropriately smooth function $\boldsymbol{u}(\boldsymbol{x}, t)$

$$
\begin{aligned}
\boldsymbol{u}_t + \mathcal{N}_{\boldsymbol{x}}[\boldsymbol{u}] &= 0, \quad \boldsymbol{x} \in D, t \in [0, T] \\
\boldsymbol{u}(\boldsymbol{x}, 0) &= h(\boldsymbol{x}), \quad \boldsymbol{x} \in D \\
\boldsymbol{u}(\boldsymbol{x}, t) &= g(\boldsymbol{x}, t), \quad \boldsymbol{t} \in [0, T], \boldsymbol{x} \in \partial D
\end{aligned}
\tag{1}
$$

where $\boldsymbol{x}$ and $t$ represent the space and time dimensions, subscripts denote the partial differentiation variables, $\mathcal{N}_{\boldsymbol{x}}[\boldsymbol{u}]$ is the nonlinear differential operator, $D$ is a subset of $\mathbb{R}^d$ s.t it has a well-defined boundary $\partial D$. Following Raissi et al. (2019), we try to approximate $\boldsymbol{u}(\boldsymbol{x}, t)$ by a deep neural network $\boldsymbol{u}_\theta(\boldsymbol{x}, t)$, and then we can define the corresponding residuals as,

$$
\mathcal{R}_{pde}(x, t) \coloneqq \partial_t \boldsymbol{u}_\theta + \mathcal{N}_{\boldsymbol{x}}[\boldsymbol{u}_\theta(\boldsymbol{x}, t)], \ \mathcal{R}_t(x) \coloneqq \boldsymbol{u}_\theta(\boldsymbol{x}, 0) - h(\boldsymbol{x}), \ \mathcal{R}_b(x, t) \coloneqq \boldsymbol{u}_\theta(\boldsymbol{x}, t) - g(\boldsymbol{x}, t)
$$

Note that the partial derivative of the neural network ($\boldsymbol{u}_\theta$) can be easily calculated using auto-differentiation (Baydin et al., 2018). The neural net is then trained on a loss function

$$
\mathcal{L}(\theta) \coloneqq \mathcal{L}_{pde}(\theta) + \mathcal{L}_t(\theta) + \mathcal{L}_b(\theta)
$$

where $\mathcal{L}_{pde}$, $\mathcal{L}_t$ and $\mathcal{L}_b$ penalize for $\mathcal{R}_{pde}$, $\mathcal{R}_t$ and $\mathcal{R}_b$ respectively for being non-zero. Typically it would take the form

$$
\mathcal{L}_{pde} = \frac{1}{N_{pde}} \sum_{i=1}^{N_{pde}} \left| \mathcal{R}_{pde}(x_r^i, t_r^i) \right|^2, \ \mathcal{L}_t = \frac{1}{N_t} \sum_{i=1}^{N_t} \left| \mathcal{R}_t(x_t^i) \right|^2, \ \mathcal{L}_b = \frac{1}{N_b} \sum_{i=1}^{N_b} \left| \mathcal{R}_b(x_b^i, t_b^i) \right|^2
$$

where $(x_r^i, t_r^i)$ denotes the collocation points, $(x_t^i)$ are the points sampled on the spatial domain for the initial loss and $(x_b^i, t_b^i)$ are the points sampled on the boundary for the boundary loss. The aim here is to train a neural net $\boldsymbol{u}_\theta$ such that $\mathcal{L}_\theta$ is as close to zero as possible.

## 2 Related Works

To the best of our knowledge the most general population risk bound for PINNs has been proven in Hu et al. (2022b), and this result applies to all linear second order PDE and it is a Rademacher complexity based bound. This bound cannot be applied to our study since Burgers' PDE is not a linear PDE. Mishra & Molinaro (2022) derived generalization bounds for PINNs, that unlike Hu et al. (2022b), explicitly depend on the trained neural net. They performed the analysis for several

---

[2]One can surmise that it might be possible to build better theory exploiting information about the blow-ups - like if the temporal location of the blow-up is known. However, it is to be noted that building theory while assuming knowledge of the location of the blow-up might be deemed unrealistic given the real-world motivations for such phenomena.

PDEs, and the "viscous scalar conservation law" being one of them, which includes the $1 + 1$-Burgers' PDE. However for testing against analytic blow-up solutions, we need such bounds at zero viscosity unlike what is considered therein, and most critically, unlike Mishra & Molinaro (2022) we keep track of the prediction error at the spatial boundary of the computational domain with respect to non-trivial functional constraints.

De Ryck et al. (2022) derived a generalization bound for Navier-Stokes PDE, which too depends on the trained neural net. We note that, in contrast to the approach presented in De Ryck et al. (2022), our method does not rely on the assumption of periodicity in boundary conditions or divergence-lessness of the true solution. These flexibilities in our setup ensure that our bound applies to known analytic cases of finite-time blow-ups for the $d + 1$-Burgers' PDE.

Notwithstanding the increasing examples of the success of PINNs, it is known that PINNs can at times fail to converge to the correct solution even for basic PDEs – as reflected in several recent studies on characterizing the "failure modes" of PINNs. Studies by Wang et al. (2021a), and more recently by Daw et al. (2023) have demonstrated that sometimes this failure can be attributed to problems associated with the loss function, specifically the uneven distribution of gradients across various components of the PINN loss. Wang et al. (2021a) attempt to address this issue by assigning specific weights to certain parts of the loss function. While Daw et al. (2022) developed a way to preferentially sample collocation points with high loss and subsequently use them for training. Krishnapriyan et al. (2021) observed a similar issue within the structure of the loss function. While not changing the PINN loss function, they introduced two techniques: "curriculum regularization" and "sequence-to-sequence learning" for PINNs to enhance their performance. In Wang et al. (2022c) PINNs have been analyzed from a neural tangent kernel perspective to suggest that PINNs suffer from "spectral-bias"(Rahaman et al., 2019) which makes it more susceptible to failure in the presence of "high frequency features" in the target function. They propose a method for improving training by assigning weights to individual components of the loss functions, aiming to mitigate the uneven convergence rates among the various loss elements.

**Notation** In the subsequent section we use $d + 1$ to represent dimensions, here $d$ is the number of spatial dimensions and 1 is always the temporal dimension. Nabla ($\nabla$) is used to represent the differential operator i.e. $(\frac{\partial}{\partial x_1}, \ldots, \frac{\partial}{\partial x_d})$. And for any real function $u$ on a domain $D$, $\|u(x)\|_{L^\infty(D)}$ will represent $\sup_{x \in D} |u(x)|$.

## 3 MAIN RESULTS

### 3.1 GENERALIZATION BOUNDS FOR THE $(d + 1)$-DIMENSIONAL BURGERS' PDE

The PDE that we consider is as follows,

$$\partial_t \boldsymbol{u} + (\boldsymbol{u} \cdot \nabla)\boldsymbol{u} = 0$$
$$\boldsymbol{u}(t = t_0) = \boldsymbol{u}_{t_0} \tag{2}$$

Here $\boldsymbol{u} : D \times [t_0, T] \to \mathbb{R}^d$ is the fluid velocity and $\boldsymbol{u}_{t_0} : D \to \mathbb{R}^d$ is the initial velocity. Then corresponding to a surrogate solution $\boldsymbol{u}_\theta$ we define the residuals as,

$$\mathcal{R}_{\text{pde}} \coloneqq \partial_t \boldsymbol{u}_\theta + (\boldsymbol{u}_\theta \cdot \nabla)\boldsymbol{u}_\theta \tag{3}$$
$$\mathcal{R}_{\text{t}} \coloneqq \boldsymbol{u}_\theta(t = t_0) - \boldsymbol{u}(t = t_0) \tag{4}$$

Corresponding to the true solution $\boldsymbol{u}$, we will define the $L^2$- risk of any surrogate solution $\boldsymbol{u}_\theta$ as,

$$\int_\Omega \|\boldsymbol{u}(\boldsymbol{x}, t) - \boldsymbol{u}_\theta(\boldsymbol{x}, t)\|_2^2 \, \mathrm{d}\boldsymbol{x} \, \mathrm{d}t$$

In the following theorem we consider $t_0 = \frac{-1}{\sqrt{2}} + \delta$ and $T = \delta$ for some $\delta > 0$. Here the spatial domain is represented by $D \subset \mathbb{R}^d$ and $\Omega$ represents the whole domain $D \times [t_0, T]$.

**Theorem 3.1.** *Let $d \in \mathbb{N}$ and $\boldsymbol{u} \in C^1(D \times [t_0, T])$ be the unique solution of the (d+1)-dimensional Burgers' equation given in equation 2. Then for any $C^1$ surrogate solution to equation 2, say $\boldsymbol{u}_\theta$, the $L^2$-risk with respect to the true solution is bounded as,*

$$\log\left(\int_\Omega \|\boldsymbol{u}(\boldsymbol{x}, t) - \boldsymbol{u}_\theta(\boldsymbol{x}, t)\|_2^2 \, \mathrm{d}\boldsymbol{x} \, \mathrm{d}t\right) \le \log\left(\frac{C_1 C_2}{4}\right) + \frac{C_1}{\sqrt{2}} \tag{5}$$

*where,*

$$
\begin{aligned}
C_1 &= d^2 \|\nabla \boldsymbol{u}_\theta\|_{L^\infty(\Omega)} \\
&\quad + 1 + d^2 \|\nabla \boldsymbol{u}\|_{L^\infty(\Omega)} \\
C_2 &= \int_D \|\mathcal{R}_t\|_2^2 \, d\boldsymbol{x} + \int_\Omega \|\mathcal{R}_{pde}\|_2^2 \, d\boldsymbol{x} \, dt + d^2 \|\nabla \boldsymbol{u}_\theta\|_{L^\infty(\Omega)} \int_\Omega \|\boldsymbol{u}_\theta\|_2^2 \, d\boldsymbol{x} \, dt \\
&\quad + d^2 \|\nabla \boldsymbol{u}\|_{L^\infty(\Omega)} \int_\Omega \|\boldsymbol{u}\|_2^2 \, d\boldsymbol{x} \, dt
\end{aligned}
$$

The theorem above has been proved in Appendix A.1 We note that the bound presented in equation 5 does not make any assumptions about the existence of a blow-up in the solution and its applicable to all solutions that have continuous first derivatives however large, as would be true for the situations very close to blow-up as we would consider. Also, we note that the bound in De Ryck et al. (2022) makes assumptions (as was reviewed in Section 2) which (even if set to zero pressure) prevent it from being directly applicable to the setup above which can capture analytic solutions arbitrarily close to finite-time blow-up.

Secondly, note that these bounds are not dependent on the details of the loss function that might eventually be used in the training to obtained the $\boldsymbol{u}_\theta$. In that sense such a bound is more universal than usual generalization bounds which depend on the loss.

Lastly, note that the inequality proven in Theorem 3.1 bounds the distance of the true solution from a PINN solution in terms of (a) norms of the true solution and (b) various integrals of the found solution like its norms and unsupervised risks on the computation domain. Hence this is not like usual generalization bounds that get proven in deep-learning theory literature where the LHS is the population risk and RHS is upperbounding that by a function that is entirely computable from knowing the training data and the trained net. Being in the setup of solving PDEs via nets lets us contruct such new kinds of bounds which can exploit knowledge of the true PDE solution.

While Theorem 3.1 is applicable to Burgers' equations in any dimensions, it becomes computationally very expensive to compute the bound in higher dimensions. Therefore, in order to better our intuitive understanding, we separately analyze the case of $d = 1$, in the upcoming Section 3.2. Furthermore, the RHS of (5) only sees the errors at the initial time and in the space-time bulk. In general dimensions it is rather complicated to demonstrate that being able to measure the boundary risks of the surrogate solution can be leveraged to get stronger generalization bounds. But this can be transparently kept track of in the $d = 1$ case - as we will demonstrate now for a specific case with finite-time blow-up. Along the way, it will also be demonstrated that the bounds possible in one dimension - are "stable" in a precise sense as will be explained after the following theorem.

### 3.2 Generalization Bounds for a Finite-Time Blow-Up Scenario with (1+1)-dimensional Burgers' PDE

For $u : [-1, 1] \times [t_0, T] \to \mathbb{R}$ being at least once continuously differentiable in each of its variables we consider a Burgers's PDE as follows on the space domain being $[-1, 1]$ and the two limits of the time domain being specified as $t_0 = -1 + \delta$ and $T = \delta$ for any $\delta > 0$,

$$
\begin{aligned}
u_t + u u_x &= 0 \\
u(x, -1 + \delta) &= \frac{x}{-2 + \delta} \\
u(-1, t) = \frac{1}{1 - t} \;&; \; u(1, t) = \frac{1}{t - 1}
\end{aligned}
\tag{6}
$$

We note that in the setup for Burger's PDE being solved by neural nets that was analyzed in the pioneering work in Mishra & Molinaro (2022), the same amount of information was assumed to be known i.e the PDE, an initial condition and boundary conditions at the spatial boundaries. However in here, the values we choose for the above constraints are non-trivial and designed to cater to a known solution for this PDE i.e $u = \frac{x}{t-1}$ which blows up at $t = 1$.

For any $C^1$ surrogate solution to the above, say $u_\theta$ its residuals can be written as,

$$\mathcal{R}_{int,\theta}(x,t) \coloneqq \partial_t(u_\theta(x,t)) + \partial_x \frac{u_\theta^2(x,t)}{2} \tag{7}$$

$$\mathcal{R}_{tb,\theta}(x) \coloneqq u_\theta(x, -1+\delta) - \frac{x}{-2+\delta} \tag{8}$$

$$(\mathcal{R}_{sb,-1,\theta}(t), \mathcal{R}_{sb,1,\theta}(t)) \coloneqq \left( u_\theta(-1,t) - \frac{1}{1-t}, \ u_\theta(1,t) - \frac{1}{t-1} \right) \tag{9}$$

We define the $L^2$−risk of $u_\theta$ with respect to the true solution $u$ of equation 6 as,

$$\mathcal{E}_G(u_\theta) \coloneqq \left( \int_{-1+\delta}^{\delta} \int_{-1}^{1} |u(x,t) - u_\theta(x,t)|^2 \ dxdt \right)^{\frac{1}{2}} \tag{10}$$

**Theorem 3.2.** *Let $u \in C^k((-1+\delta,\delta) \times (-1,1))$ be the unique solution of the one dimensional Burgers' PDE in equation 6, for any $k \geq 1$. Then for any surrogate solution for the same PDE, say $u^* \coloneqq u_{\theta^*}$ its risk as defined in equation 10 is bounded as,*

$$
\begin{aligned}
\mathcal{E}_G^2 \leq & \left[1 + Ce^C\right] \left[ \int_{-1}^{1} \mathcal{R}_{tb,\theta^*}(x)dx + 2C_{2b}\left( \int_{-1+\delta}^{\delta} \mathcal{R}_{sb,-1,\theta^*}^2(t)dt + \int_{-1+\delta}^{\delta} \mathcal{R}_{sb,1,\theta^*}^2(t)dt \right) \right. \\
& + 2C_{1b}\left( \left( \int_{-1+\delta}^{\delta} \mathcal{R}_{sb,-1,\theta^*}^2(t)dt \right)^{\frac{1}{2}} + \left( \int_{-1+\delta}^{\delta} \mathcal{R}_{sb,1,\theta^*}^2(t)dt \right)^{\frac{1}{2}} \right) \\
& \left. + \int_{-1+\delta}^{\delta} \int_{-1}^{1} \mathcal{R}_{int,\theta^*}^2(x,t)dxdt \right]
\end{aligned}
\tag{11}
$$

*where $C = 1 + 2C_{u_x}$, with $C_{u_x} = \|u_x\|_{L^\infty((-1+\delta,\delta)\times(-1,1))} = \left\| \frac{1}{t-1} \right\|_{L^\infty([-1+\delta,\delta])} = \frac{1}{1-\delta}$ and*

$$C_{1b} = \|u(1,t)\|_{L^\infty([-1+\delta,\delta])}^2 = \left\| \frac{1}{1-t} \right\|_{L^\infty([-1+\delta,\delta])}^2 = \frac{1}{(1-\delta)^2}$$

$$C_{2b} = \|u_{\theta^*}(1,t)\|_{L^\infty([-1+\delta,\delta])} + \frac{3}{2}\left\| \frac{1}{t-1} \right\|_{L^\infty([-1+\delta,\delta])} = \|u_{\theta^*}(1,t)\|_{L^\infty([-1+\delta,\delta])} + \frac{3}{2}\left( \frac{1}{1-\delta} \right) \tag{12}$$

The theorem above has been proved in Appendix A.3. Note that the RHS of equation 11 is evaluatable without exactly knowing the eact true solution – the constants in equation 11 only requires some knowledge of the supremum value of $u$ at the spatial boundaries and the behaviour of the first order partial derivative of $u$.

Most importantly, Theorem 3.2 shows that despite the setting here being of proximity to finite-time blow-up, the naturally motivated PINN risk in this case [3] is "$(L_2, L_2, L_2, L_2)$-stable"[4] in the precise sense as defined in Wang et al. (2022a). This stability property being true implies that if the PINN risk of the solution obtained is measured to be $\mathcal{O}(\epsilon)$ then it would directly imply that the $L_2$-risk with respect to the true solution (10) is also $\mathcal{O}(\epsilon)$. And this would be determinable *without having to know the true solution at test time.*

In Appendix A.4 we apply quadrature rules on (11) and show a version of the above bound which makes the sample size dependency of the bound more explicit.

## 4 EXPERIMENTS

Our experiments are designed to demonstrate the efficacy of the generalization error bounds discussed in Section 3 in the vicinity of finite-time blow-ups happening in our use cases. Towards motivating the novelty of our setup we give a brief overview of how demonstrations of deep-learning generalization bounds have been done in the recent past.

---

[3] PINN risk is defined as $\mathbb{E}[|\mathcal{R}_{int,\theta}(x,t)|^2] + \mathbb{E}[|\mathcal{R}_{tb,\theta}|^2] + \mathbb{E}[|\mathcal{R}_{sb,-1,\theta}|^2] + \mathbb{E}[|\mathcal{R}_{sb,1,\theta}|^2]$

[4] Suppose $Z_1, Z_2, Z_3, Z_4$ are four Banach spaces, a PDE defined by (1) is $Z_1, Z_2, Z_3, Z_4$-stable, if $\|u_\theta(x,t) - u(x,t)\|_{Z4} = \mathcal{O}(\|\partial_t u_\theta + \mathcal{N}_x[u_\theta(x,t)]\|_{Z_1} + \|u_\theta(x,0) - h(x)\|_{Z_2} + \|u_\theta(x,t) - g(x,t)\|_{Z_3})$ as $\|\partial_t u_\theta + \mathcal{N}_x[u_\theta(x,t)]\|_{Z_1}, \|u_\theta(x,0) - h(x)\|_{Z_2}, \|u_\theta(x,t) - g(x,t)\|_{Z_3} \to 0$ for any $u_\theta$

In the thought-provoking paper Dziugaite & Roy (2017) the authors had computed their bounds for 2-layer neural nets at various widths to show the non-vacuous nature of their bounds. But these bounds are not applicable to any single neural net but to an expected neural net sampled from a specified distribution. Inspired by these experiments, works like Neyshabur et al. (2017) and Mukherjee (2020) perform a de-randomized PAC-Bayes analysis on the generalization error of neural nets - which can be evaluated on any given net.

In works such as Neyshabur et al. (2018) we see a bound based on Rademacher analysis of the generalization error and the experiments were performed for depth-2 nets at different widths to show the decreasing nature of their bound with increasing width – a very rare property to be true for uniform convergence based bounds. It is important to point out that the training data is kept fixed while changing the width of the neural net in Dziugaite & Roy (2017) and Neyshabur et al. (2018).

In Arora et al. (2018) the authors instantiated a way to do compression of nets and computed the bounds on a compressed version of the original net. More recently in Muthukumar & Sulam (2023) the authors incorporated the sparsity of a neural net alongside the PAC-Bayed analysis to get a better bound for the generalization error. In their experiments, they vary the data size while keeping the neural net fixed and fortuitously the bound becomes non-vacuous for a certain width of the net.

In this work, we investigate if theory can capture the performance of PINNs near a finite-time blow-up and if larger neural nets can better capture the nature of generalization error close to the blow-up. To this end, in contrast to the previous literature cited above, we keep the neural net fixed and vary the domain of the PDE. More specifically, progressively we choose time-domains arbitrarily close to the finite-time blow-up and test the theory at that difficult edge.

### 4.1 THE FINITE-TIME BLOW-UP CASE OF (1+1)-DIMENSIONAL BURGERS PDE FROM SECTION 3.2

The neural networks we use here have a depth of 6 layers, and we experiment at two distinct uniform widths of 30 and 300 neurons. For training, we use full-batch Adam optimizer for $100,000$ iterations and a learning rate of $10^{-4}$. We subsequently select the model with the lowest training error for further analysis. In Figures 1a and 1b we see that the LHS and the RHS of equation 11 measured on

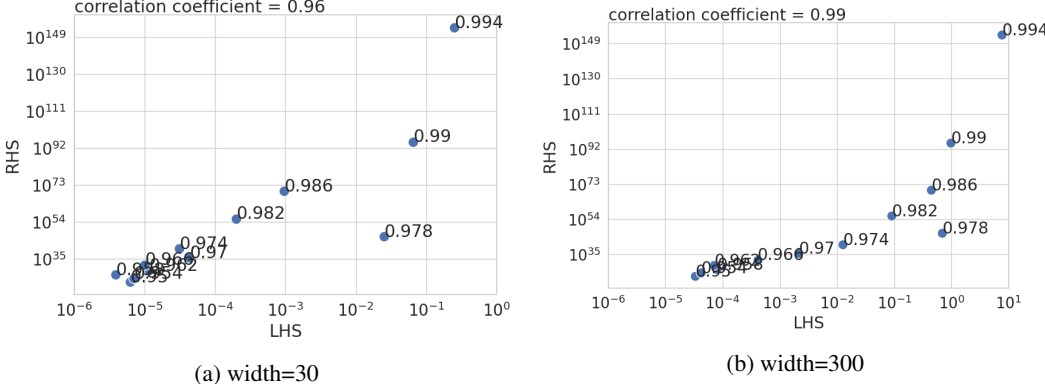

(a) width=30

(b) width=300

Figure 1: Demonstration of the presence of high correlation between the LHS (the true risk) and the RHS (and the derived bound) of equation (11) in Theorem 3.2 over PDE setups increasingly close to the singularity. Each experiment is labeled with the value of $\delta$ in the setup of equation 6 that it corresponds to.

the trained models is such that the correlation is very high (~ 1) over multiple values of the proximity parameter – up to being very close to the blow-up point. We also note that the correlation increases with the width of the neural net, a desirable phenomenon that our bound does capture – albeit implicitly. In Figure 3 in the appendix, we illustrate that the upper-bound derived in Theorem 3.2 does indeed fall over a reasonable range of widths at a fixed $\delta$. The mean and the standard deviations plotted therein are obtained over six iterations of the experiment at different random seeds.

## 4.2 Testing Against a (2+1)-dimensional Exact Burgers' Solution with Finite-Time Blow-Up

From (Biazar & Aminikhah, 2009) we know that there is an exact finite-time blow-up solution for Burgers' PDE in equation 2 for the case of $d = 2$,

$$u_1 = \frac{x_1 + x_2 - 2x_1 t}{1 - 2t^2}, \; u_2 = \frac{x_1 - x_2 - 2x_2 t}{1 - 2t^2}$$

where $u_i$ denotes the $i^{\text{th}}$ component of the velocity being solved for. Note that at $t = 0$, both the above velocities are smooth while they eventually develop singularities at $t = \frac{1}{\sqrt{2}}$ - as is the expected hallmark of non-trivial finite-time blow-up solutions of PDEs. Also note that this singularity is more difficult to solve for since it is blowing up as $\mathcal{O}(\frac{1}{t^2})$ as compared to the $\mathcal{O}(\frac{1}{t})$ blow-up in the previous section in one dimension.

We set ourselves to the task of solving for this on a sequence of computational domains $x_1, x_2 \in [0, 1]$ and $t \in [-\frac{1}{\sqrt{2}} + \delta, \delta]$ where $\delta \in [0, \frac{1}{\sqrt{2}})$. Hence we have a sequence of PDEs to solve for – parameterized by $\delta$ and larger $\delta$s getting close to the blow-up. Let $g_{x_1,0}(x_2, t)$ and $g_{x_1,1}(x_2, t)$ be the boundary conditions for $u_1$ at $x_1 = 0$ & 1. Let $g_{x_2,0}(x_1, t)$ and $g_{x_2,1}(x_1, t)$ be the boundary conditions for $u_2$ at $x_2 = 0$ & 1 and $u_{1,t_0}$ and $u_{2,t_0}$ with $t_0 = -\frac{1}{\sqrt{2}} + \delta$ be the initial conditions for the two components of the velocity field. Hence the PDE we seek to solve is,

$$\left\{ \; \boldsymbol{u}_t + (\boldsymbol{u} \cdot \nabla) \boldsymbol{u} = 0 \; \right. , \left\{ \begin{array}{l} u_{1,t_0} = \frac{(1+\sqrt{2}-2\delta)x_1 + x_2}{2\delta(\sqrt{2}-\delta)} \\ u_{2,t_0} = \frac{x_1 - (1-\sqrt{2}+2\delta)x_2}{2\delta(\sqrt{2}-\delta)} \end{array} \right. , \left\{ \begin{array}{l} g_{x_1,0}(x_2, t) := u_1(x_1 = 0) = \frac{x_2}{1-2\cdot t^2} \\ g_{x_1,1}(x_2, t) := u_1(x_1 = 1) = \frac{1+x_2-2\cdot t}{1-2\cdot t^2} \\ g_{x_2,0}(x_1, t) := u_2(x_2 = 0) = \frac{x_1}{1-2\cdot t^2} \\ g_{x_2,1}(x_1, t) := u_2(x_2 = 1) = \frac{x_1-1-2\cdot t}{1-2\cdot t^2} \end{array} \right. \tag{13}$$

Let $\mathcal{N} : \mathbb{R}^3 \to \mathbb{R}^2$ be the neural net to be trained, with output coordinates labeled as $(\mathcal{N}_{u_1}, \mathcal{N}_{u_2})$. Using this net we define the neural surrogates for solving the above PDE as,

$$u_{1,\theta} := \mathcal{N}_{u_1}(x_1, x_2, t) \; u_{2,\theta} := \mathcal{N}_{u_2}(x_1, x_2, t)$$

Correspondingly we define the PDE population risk, $\mathcal{R}_{pde}$ as,

$$\mathcal{R}_{pde} = \|\partial_t \boldsymbol{u}_\theta + \boldsymbol{u}_\theta \cdot \nabla \boldsymbol{u}_\theta\|^2_{[0,1]^2 \times [-\frac{1}{\sqrt{2}}+\delta,\delta],\nu_1} \tag{14}$$

In the above $\boldsymbol{u}_\theta = (u_{1,\theta}, u_{2,\theta})$ and $\nu_1$ is a measure on the whose space-time domain $[0, 1]^2 \times [-\frac{1}{\sqrt{2}} + \delta, \delta]$. Corresponding to a measure $\nu_2$ on $[0, 1] \times [-\frac{1}{\sqrt{2}} + \delta, \delta]$ (first interval being space and the later being time), we define $\mathcal{R}_{s,0}$ and $\mathcal{R}_{s,1}$ corresponding to violation of the boundary conditions,

$$\mathcal{R}_{s,0} = \|u_{1,\theta} - g_{x_1,0}(x_2, t)\|^2_{\{0\} \times [0,1] \times [-\frac{1}{\sqrt{2}}+\delta,\delta],\nu_2} + \|u_{2,\theta} - g_{x_2,0}(x_1, t)\|^2_{[0,1] \times \{0\} \times [-\frac{1}{\sqrt{2}}+\delta,\delta],\nu_2}$$

$$\mathcal{R}_{s,1} = \|u_{1,\theta} - g_{x_1,1}(x_2, t)\|^2_{\{1\} \times [0,1] \times [-\frac{1}{\sqrt{2}}+\delta,\delta],\nu_2} + \|u_{2,\theta} - g_{x_2,1}(x_1, t)\|^2_{[0,1] \times \{1\} \times [-\frac{1}{\sqrt{2}}+\delta,\delta],\nu_2}$$

$$\tag{15}$$

For a choice of measure $\nu_3$ on the spatial volume $[0, 1]^2$ we define $\mathcal{R}_t$ corresponding to the violation of initial conditions $\boldsymbol{u}_{t_0} = (u_1(t_0), u_2(t_0))$,

$$\mathcal{R}_t = \|\boldsymbol{u}_\theta - \boldsymbol{u}_{t_0}\|^2_{[0,1]^2, t=t_0, \nu_3} \tag{16}$$

Thus the population risk we are looking to minimize is, $\mathcal{R} = \mathcal{R}_{pde} + \mathcal{R}_{s,0} + \mathcal{R}_{s,1} + \mathcal{R}_t$

We note that for the exact solution given above the constants in Theorem 3.1 evaluate to,

$$C_1 = 2^2 \|\nabla \boldsymbol{u}_\theta\|_{L^\infty(\Omega)}$$
$$+ 1 + 2^2 \max_{t = \frac{1}{\sqrt{2}} + \delta, \delta} \left\{ \left| \frac{1-2t}{1-2t^2} \right| + \left| \frac{1}{1-2t^2} \right|, \left| \frac{1}{1-2t^2} \right| + \left| \frac{1+2t}{1-2t^2} \right| \right\}$$

$$C_2 = \int_D \|\mathcal{R}_t\|^2_2 \, \mathrm{d}\boldsymbol{x} + \int_{\tilde{\Omega}} \|\mathcal{R}_{pde}\|^2_2 \, \mathrm{d}\boldsymbol{x} \, \mathrm{d}t + 2^2 \|\nabla \boldsymbol{u}_\theta\|_{L^\infty(\Omega)} \int_{\tilde{\Omega}} \|\boldsymbol{u}_\theta\|^2_2 \, \mathrm{d}\boldsymbol{x} \, \mathrm{d}t$$
$$+ 2^2 \max_{t = \frac{1}{\sqrt{2}} + \delta, \delta} \left\{ \left| \frac{1-2t}{1-2t^2} \right| + \left| \frac{1}{1-2t^2} \right|, \left| \frac{1}{1-2t^2} \right| + \left| \frac{1+2t}{1-2t^2} \right| \right\} \int_{\tilde{\Omega}} \|\boldsymbol{u}\|^2_2 \, \mathrm{d}\boldsymbol{x} \, \mathrm{d}t$$

$$= \int_D \|\mathcal{R}_t\|_2^2 \, \mathrm{d}\boldsymbol{x} + \int_{\tilde{\Omega}} \|\mathcal{R}_{pde}\|_2^2 \, \mathrm{d}\boldsymbol{x} \, \mathrm{d}t + 2^2 \|\nabla \boldsymbol{u}_\theta\|_{L^\infty(\Omega)} \int_{\tilde{\Omega}} \|\boldsymbol{u}_\theta\|_2^2 \, \mathrm{d}\boldsymbol{x} \, \mathrm{d}t$$

$$+ 2^2 \max_{t=\frac{1}{\sqrt{2}}+\delta,\delta} \left\{ \left| \frac{1-2t}{1-2t^2} \right| + \left| \frac{1}{1-2t^2} \right|, \left| \frac{1}{1-2t^2} \right| + \left| \frac{1+2t}{1-2t^2} \right| \right\} \left[ \frac{11t-7}{12(1-2t^2)} + \frac{5t+1}{12(1-2t^2)} \right]_{t=\frac{1}{\sqrt{2}}+\delta}^{\delta}$$

(a) width=30             (b) width=100

Figure 2: These plots show the behaviour of LHS (the true risk) and RHS (the derived bound) of equation (5) in Theorem 3.1 for different values of the $\delta$ parameter that quantifies proximity to the blow-up point. In the left plot each point is marked with the value of the $\delta$ at which the experiment is done and the right figure, for clarity, this is marked only for experiments at $\delta > \frac{1}{2}$.

In figure 5 we see the true risk and the derived bound in Theorem 3.1 for depth 6 neural nets obtained by training on the above loss. The experiments show that the insight from the previous demonstration continues to hold and more vividly so. Here, for the experiments at low width (30) the correlation stays around $0.50$ until only $\delta = 0.307$, and beyond that it decreases rapidly. However, for experiments at width 100 the correlation remains close to $0.80$ for $\delta$ much closer to the blow-up i.e at $\frac{1}{\sqrt{2}}$.

## 5    CONCLUDING DISCUSSIONS

In this work we have taken some of the first-of-its kind steps to initiate research into understanding the ability of neural nets to solve PDEs at the edge of finite-time blow-up. Our work suggests a number of exciting directions of future research. Firstly, more sophisticated modifications to the PINN formalism could be found to solve PDEs specifically near finite-time blow-ups.

Secondly, we note that it remains an open question to establish if there is any PINN risk for the $d + 1$-dimensional Burgers, for $d > 1$, that is stable by the condition stated in Wang et al. (2022a), as was shown to be true in our $1 + 1$-dimensional Burgers in Theorem 3.2.

In Luo & Hou (2014b) the authors had given numerical studies to suggest that 3D incompressible Euler PDEs can develop finite-time singularities from smooth initial conditions for the fluid velocity. For their setup of axisymmetric fluid flow they conjectured a simplified model for the resultant flow near the outer boundary of the cylinder. Self-similar finite-time blow-ups for this model's solutions were rigorously established in Chen et al. (2022) - and it was shown that an estimate of its blowup-exponent is very close to the measured values of the 3D Euler PDE. In the seminal paper Elgindi (2021) it was shown that the unique local solution to 3D incompressible Euler PDEs can develop finite-time singularities despite starting from a divergence-free and odd initial velocity in $\mathbb{C}^{1,\alpha}$ and initial vorticity bounded as $\sim \frac{1}{1+\|x\|^\alpha}$. This breakthrough was built upon to prove the existence of finite time singularity in 2D Boussinesq PDE in Chen & Hou (2021).

Luo & Hou (2014a) highlighted the association between blow-ups in 3D Euler and 2D Boussinesq PDEs. In Wang et al. (2022d), the authors investigated the ability for PINNs to detect the occurrence of self-similar blow-ups in 2D Boussinesq PDE. A critical feature of this experiment was its use of the unconventional regularizer on the gradients of the neural surrogate with respect to its inputs. In light of this, we posit that a very interesting direction of research would be to investigate if a theoretical analysis of the risk bound for such losses can be used as a method of detection of the blow-up.

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

# A  PROOFS FOR THE MAIN THEOREMS

## A.1  PROOF OF THEOREM 3.1

Let $\boldsymbol{u}$ be the actual solution of the (d+1)-dimensional Burgers' PDE and $\boldsymbol{u}_\theta$ the predicted solution by the PINN with parameters $\theta$. Let's define

$$f(\boldsymbol{u}) \coloneqq \frac{\|\boldsymbol{u}\|_2^2}{2}$$

$$\hat{\boldsymbol{u}} \coloneqq \boldsymbol{u}_\theta - \boldsymbol{u}$$

Then we can write the (d+1)-dimenional Burgers' and it's residual as

$$\partial_t \boldsymbol{u} + (\boldsymbol{u} \cdot \nabla)\boldsymbol{u} = 0 \tag{17}$$

$$\mathcal{R}_{\mathrm{pde}} \coloneqq \partial_t \boldsymbol{u}_\theta + (\boldsymbol{u}_\theta \cdot \nabla)\boldsymbol{u}_\theta \tag{18}$$

Now multiplying (18) with $\boldsymbol{u}_\theta$ on both sides we get

$$\boldsymbol{u}_\theta \cdot \mathcal{R}_{\mathrm{pde}} = \partial_t f(\boldsymbol{u}_\theta) + \boldsymbol{u}_\theta \cdot \nabla f(\boldsymbol{u}_\theta) \tag{19}$$

Similarly, multiplying both sides of (17) with $\boldsymbol{u}$ we get

$$\partial_t f(\boldsymbol{u}) + \boldsymbol{u} \cdot \nabla f(\boldsymbol{u}) = 0 \tag{20}$$

Some calculation on (17) and (18) yields

$$\boldsymbol{u} \cdot \left[(\partial_t \boldsymbol{u}_\theta + (\boldsymbol{u}_\theta \cdot \nabla)\boldsymbol{u}_\theta - \mathcal{R}_{\mathrm{pde}}) - (\partial_t \boldsymbol{u} + (\boldsymbol{u} \cdot \nabla)\boldsymbol{u})\right] = -\hat{\boldsymbol{u}} \cdot \left[\partial_t \boldsymbol{u} + (\boldsymbol{u} \cdot \nabla)\boldsymbol{u}\right]$$

$$\implies \partial_t(\boldsymbol{u} \cdot \hat{\boldsymbol{u}}) + \boldsymbol{u} \cdot ((\boldsymbol{u}_\theta \cdot \nabla)\boldsymbol{u}_\theta) - \boldsymbol{u} \cdot ((\boldsymbol{u} \cdot \nabla)\boldsymbol{u}) - \boldsymbol{u} \cdot \mathcal{R}_{\mathrm{pde}} = -\hat{\boldsymbol{u}} \cdot ((\boldsymbol{u} \cdot \nabla)\boldsymbol{u})$$

$$\implies \partial_t(\boldsymbol{u} \cdot \hat{\boldsymbol{u}}) + \boldsymbol{u} \cdot ((\boldsymbol{u}_\theta \cdot \nabla)\boldsymbol{u}_\theta) - \boldsymbol{u} \cdot \nabla f(\boldsymbol{u}) - \boldsymbol{u} \cdot \mathcal{R}_{\mathrm{pde}} = -\hat{\boldsymbol{u}} \cdot ((\boldsymbol{u} \cdot \nabla)\boldsymbol{u}) \tag{21}$$

Let

$$S \coloneqq \frac{1}{2}\hat{\boldsymbol{u}} \cdot \hat{\boldsymbol{u}} \tag{22}$$

$$\implies \partial_t S = \partial_t f(\boldsymbol{u}_\theta) - \partial_t f(\boldsymbol{u}) - \partial_t(\boldsymbol{u} \cdot \hat{\boldsymbol{u}})$$

$$= \left[\boldsymbol{u}_\theta \cdot \mathcal{R}_{\mathrm{pde}} - \boldsymbol{u}_\theta \cdot \nabla f(\boldsymbol{u}_\theta)\right] + \left[\boldsymbol{u} \cdot \nabla f(\boldsymbol{u})\right]$$

$$\quad - \left[-\boldsymbol{u} \cdot ((\boldsymbol{u}_\theta \cdot \nabla)\boldsymbol{u}_\theta) + \boldsymbol{u} \cdot \nabla f(\boldsymbol{u}) + \boldsymbol{u} \cdot \mathcal{R}_{\mathrm{pde}} - \hat{\boldsymbol{u}} \cdot ((\boldsymbol{u} \cdot \nabla)\boldsymbol{u})\right]$$

$$= \hat{\boldsymbol{u}} \cdot \mathcal{R}_{\mathrm{pde}} - \boldsymbol{u}_\theta \cdot \nabla f(\boldsymbol{u}_\theta) + \boldsymbol{u} \cdot ((\boldsymbol{u}_\theta \cdot \nabla)\boldsymbol{u}_\theta) + \hat{\boldsymbol{u}} \cdot ((\boldsymbol{u} \cdot \nabla)\boldsymbol{u})$$

$$= \hat{\boldsymbol{u}} \cdot \mathcal{R}_{\mathrm{pde}} + \hat{\boldsymbol{u}} \cdot ((\boldsymbol{u} \cdot \nabla)\boldsymbol{u} - (\boldsymbol{u}_\theta \cdot \nabla)\boldsymbol{u}_\theta)$$

here we represent the spatial domain $[0,1] \times [0,1]$ by $D$ and use $\Omega$ to represent the $D \times [-\frac{1}{\sqrt{2}} + \delta, \delta]$. We then define

$$\mathcal{T} \coloneqq \hat{\boldsymbol{u}} \cdot ((\boldsymbol{u} \cdot \nabla)\boldsymbol{u}) \tag{23}$$

$$\tilde{H} \coloneqq \hat{\boldsymbol{u}} \cdot ((\boldsymbol{u}_\theta \cdot \nabla)\boldsymbol{u}_\theta) \tag{24}$$

And this leads to,

$$\partial_t S + \tilde{H} = \hat{\boldsymbol{u}} \cdot \mathcal{R}_{\mathrm{pde}} + \mathcal{T} \tag{25}$$

Thus we have the inequalities,

$$\int_D \partial_t \|\hat{\boldsymbol{u}}\|_2^2 \, \mathrm{d}\boldsymbol{x} \le \int_D \|\hat{\boldsymbol{u}}\|_2^2 \, \mathrm{d}\boldsymbol{x} + \int_D \|\mathcal{R}_{pde}\|_2^2 \, \mathrm{d}\boldsymbol{x}$$

$$\quad + 2\int_D \hat{\boldsymbol{u}} \cdot ((\boldsymbol{u} \cdot \nabla)\boldsymbol{u}) \, \mathrm{d}\boldsymbol{x} - 2\int_D \hat{\boldsymbol{u}} \cdot ((\boldsymbol{u}_\theta \cdot \nabla)\boldsymbol{u}_\theta) \, \mathrm{d}\boldsymbol{x}$$

$$\quad \le \int_D \|\hat{\boldsymbol{u}}\|_2^2 \, \mathrm{d}\boldsymbol{x} + \int_D \|\mathcal{R}_{pde}\|_2^2 \, \mathrm{d}\boldsymbol{x}$$

$$\quad + 2d^2\|\nabla\boldsymbol{u}\|_{L^\infty(\Omega)} \int_D \|\boldsymbol{u}\|_2 \|\hat{\boldsymbol{u}}\|_2 \, \mathrm{d}\boldsymbol{x} + 2d^2\|\nabla\boldsymbol{u}_\theta\|_{L^\infty(\Omega)} \int_D \|\boldsymbol{u}_\theta\|_2 \|\hat{\boldsymbol{u}}\|_2 \, \mathrm{d}\boldsymbol{x}$$

$$\quad \le \int_D \|\hat{\boldsymbol{u}}\|_2^2 \, \mathrm{d}\boldsymbol{x} + \int_D \|\mathcal{R}_{pde}\|_2^2 \, \mathrm{d}\boldsymbol{x}$$

$$\quad + d^2\|\nabla\boldsymbol{u}\|_{L^\infty(\Omega)} \int_D \left[\|\boldsymbol{u}\|_2^2 + \|\hat{\boldsymbol{u}}\|_2^2\right] \mathrm{d}\boldsymbol{x} + d^2\|\nabla\boldsymbol{u}_\theta\|_{L^\infty(\Omega)} \int_D \left[\|\boldsymbol{u}_\theta\|_2^2 + \|\hat{\boldsymbol{u}}\|_2^2\right] \mathrm{d}\boldsymbol{x}$$

$$\leq \left[1 + d^2 \|\nabla \boldsymbol{u}\|_{L^\infty(\Omega)} + d^2 \|\nabla \boldsymbol{u}_\theta\|_{L^\infty(\Omega)}\right] \int_D \|\hat{\boldsymbol{u}}\|_2^2 \, \mathrm{d}\boldsymbol{x}$$

$$+ \int_D \|\mathcal{R}_{pde}\|_2^2 \, \mathrm{d}\boldsymbol{x} + d^2 \|\nabla \boldsymbol{u}\|_{L^\infty(\Omega)} \int_D \|\boldsymbol{u}\|_2^2 \, \mathrm{d}\boldsymbol{x} + d^2 \|\nabla \boldsymbol{u}_\theta\|_{L^\infty(\Omega)} \int_D \|\boldsymbol{u}_\theta\|_2^2 \, \mathrm{d}\boldsymbol{x}$$

$$\leq C_1 \int_D \|\hat{\boldsymbol{u}}\|_2^2 \, \mathrm{d}\boldsymbol{x} + \int_D \|\mathcal{R}_{pde}\|_2^2 \, \mathrm{d}\boldsymbol{x}$$

$$+ d^2 \|\nabla \boldsymbol{u}\|_{L^\infty(\Omega)} \int_D \|\boldsymbol{u}\|_2^2 \, \mathrm{d}\boldsymbol{x} + d^2 \|\nabla \boldsymbol{u}_\theta\|_{L^\infty(\Omega)} \int_D \|\boldsymbol{u}_\theta\|_2^2 \, \mathrm{d}\boldsymbol{x} \tag{26}$$

where

$$C_1 = d^2 \|\nabla \boldsymbol{u}_\theta\|_{L^\infty(\Omega)}$$
$$+ 1 + d^2 \|\nabla \boldsymbol{u}\|_{L^\infty(\Omega)}$$

Lets define the domain $D \times [-\frac{1}{\sqrt{2}} + \delta, \tilde{\delta}]$ by $\tilde{\Omega}$ where $\tilde{\delta} \in [-\frac{1}{\sqrt{2}} + \delta, \delta)$.

Integrating over $\tilde{\Omega}$ we get

$$\int_{\tilde{\Omega}} \partial_t \|\hat{\boldsymbol{u}}\|_2^2 \, \mathrm{d}\boldsymbol{x} \, \mathrm{d}t \leq C_1 \int_{\tilde{\Omega}} \|\hat{\boldsymbol{u}}\|_2^2 \, \mathrm{d}\boldsymbol{x} \, \mathrm{d}t + \int_{\tilde{\Omega}} \|\mathcal{R}_{pde}\|_2^2 \, \mathrm{d}\boldsymbol{x} \, \mathrm{d}t$$

$$+ d^2 \|\nabla \boldsymbol{u}\|_{L^\infty(\Omega)} \int_{\tilde{\Omega}} \|\boldsymbol{u}\|_2^2 \, \mathrm{d}\boldsymbol{x} \, \mathrm{d}t + d^2 \|\nabla \boldsymbol{u}_\theta\|_{L^\infty(\Omega)} \int_{\tilde{\Omega}} \|\boldsymbol{u}_\theta\|_2^2 \, \mathrm{d}\boldsymbol{x} \, \mathrm{d}t$$

$$\leq \int_D \|\mathcal{R}_t\|_2^2 \, \mathrm{d}\boldsymbol{x} + C_1 \int_{\tilde{\Omega}} \|\hat{\boldsymbol{u}}\|_2^2 \, \mathrm{d}\boldsymbol{x} \, \mathrm{d}t + \int_{\tilde{\Omega}} \|\mathcal{R}_{pde}\|_2^2 \, \mathrm{d}\boldsymbol{x} \, \mathrm{d}t$$

$$+ d^2 \|\nabla \boldsymbol{u}\|_{L^\infty(\Omega)} \int_{\tilde{\Omega}} \|\boldsymbol{u}\|_2^2 \, \mathrm{d}\boldsymbol{x} \, \mathrm{d}t + d^2 \|\nabla \boldsymbol{u}_\theta\|_{L^\infty(\Omega)} \int_{\tilde{\Omega}} \|\boldsymbol{u}_\theta\|_2^2 \, \mathrm{d}\boldsymbol{x} \, \mathrm{d}t$$

$$\leq \int_D \|\mathcal{R}_t\|_2^2 \, \mathrm{d}\boldsymbol{x} + C_1 \int_\Omega \|\hat{\boldsymbol{u}}\|_2^2 \, \mathrm{d}\boldsymbol{x} \, \mathrm{d}t + \int_\Omega \|\mathcal{R}_{pde}\|_2^2 \, \mathrm{d}\boldsymbol{x} \, \mathrm{d}t$$

$$+ d^2 \|\nabla \boldsymbol{u}\|_{L^\infty(\Omega)} \int_\Omega \|\boldsymbol{u}\|_2^2 \, \mathrm{d}\boldsymbol{x} \, \mathrm{d}t + d^2 \|\nabla \boldsymbol{u}_\theta\|_{L^\infty(\Omega)} \int_\Omega \|\boldsymbol{u}_\theta\|_2^2 \, \mathrm{d}\boldsymbol{x} \, \mathrm{d}t$$

$$\leq C_1 \int_{\tilde{\Omega}} \|\hat{\boldsymbol{u}}\|_2^2 \, \mathrm{d}\boldsymbol{x} \, \mathrm{d}t + C_2 \tag{27}$$

where,

$$C_2 = \int_D \|\mathcal{R}_t\|_2^2 \, \mathrm{d}\boldsymbol{x} + \int_{\tilde{\Omega}} \|\mathcal{R}_{pde}\|_2^2 \, \mathrm{d}\boldsymbol{x} \, \mathrm{d}t + d^2 \|\nabla \boldsymbol{u}_\theta\|_{L^\infty(\Omega)} \int_{\tilde{\Omega}} \|\boldsymbol{u}_\theta\|_2^2 \, \mathrm{d}\boldsymbol{x} \, \mathrm{d}t$$
$$+ d^2 \|\nabla \boldsymbol{u}\|_{L^\infty(\Omega)} \int_{\tilde{\Omega}} \|\boldsymbol{u}\|_2^2 \, \mathrm{d}\boldsymbol{x} \, \mathrm{d}t$$

Applying Gronwall's inequality on (27) we get

$$\int_D \left\|\hat{\boldsymbol{u}}(\boldsymbol{x}, \tilde{\delta})\right\|_2^2 \, \mathrm{d}\boldsymbol{x} \leq C_2 + \int_{-\frac{1}{\sqrt{2}}+\delta}^{\tilde{\delta}} C_2 C_1 e^{\int_t^\delta C_1 \mathrm{d}s} \, \mathrm{d}t \leq C_2 \left[1 + \int_{-\frac{1}{\sqrt{2}}+\delta}^{\tilde{\delta}} C_1 e^{\frac{C_1}{\sqrt{2}}} \, \mathrm{d}t\right] \tag{28}$$

Integrating (28) over $\mathrm{d}\tilde{\delta}$ we get

$$\int_\Omega \left\|\hat{\boldsymbol{u}}(\boldsymbol{x}, \tilde{\delta})\right\|_2^2 \, \mathrm{d}\boldsymbol{x} \, \mathrm{d}\tilde{\delta} \leq C_2 \int_{\frac{-1}{\sqrt{2}}+\delta}^{\delta} \left[1 + \int_{-\frac{1}{\sqrt{2}}+\delta}^{\tilde{\delta}} C_1 e^{\frac{C_1}{\sqrt{2}}} \, \mathrm{d}t\right] \mathrm{d}\tilde{\delta}$$

$$\leq C_2 \left[\frac{-1}{\sqrt{2}} + \int_{\frac{-1}{\sqrt{2}}+\delta}^{\delta} \int_{-\frac{1}{\sqrt{2}}+\delta}^{\tilde{\delta}} C_1 e^{\frac{C_1}{\sqrt{2}}} \, \mathrm{d}t \, \mathrm{d}\tilde{\delta}\right]$$

$$\leq C_2 \left[\frac{-1}{\sqrt{2}} + \frac{C_1}{4} e^{\frac{C_1}{\sqrt{2}}}\right]$$

$$\implies \log\left(\int_\Omega \left\|\hat{\boldsymbol{u}}(\boldsymbol{x}, \tilde{\delta})\right\|_2^2 \, \mathrm{d}\boldsymbol{x} \, \mathrm{d}\tilde{\delta}\right) \leq \log\left(\frac{C_1 C_2}{4}\right) + \frac{C_1}{\sqrt{2}} \tag{29}$$

## A.2 USEFUL LEMMAS

**Lemma A.1.**

$$
\begin{aligned}
\int_D p \cdot ((q \cdot \nabla) r) dx &= \int_D \left[ \sum_{i=1}^d p_i (q \cdot \nabla r_i) \right] dx \\
&\leq \int_D \left[ \sum_{i=1}^d \|p\|_2 (\|q\|_2 \|\nabla r_i\|_2) \right] dx \\
&\leq \int_D \|p\|_2 \|q\|_2 \left[ \sum_{i=1}^d \|\nabla r_i\|_2 \right] dx \\
&\leq \int_D \|p\|_2 \|q\|_2 \left[ \sum_{i=1}^d d \|\nabla r\|_{L^\infty(\Omega)} \right] dx \\
&\leq d^2 \|\nabla r\|_{L^\infty(\Omega)} \int_D \|p\|_2 \|q\|_2 dx
\end{aligned}
\tag{30}
$$

### A.3 Proof of Theorem 3.2

*Proof.* We define

$$f(u) = \frac{u^2}{2}$$

which means the first equation in (6) can be written as

$$u_t + f(u)_x = 0 \tag{31}$$

Then we define the entropy flux function as

$$\mathcal{Q}(u) = \int_a^u s f'(s) ds \quad \text{for any } a \in \mathbb{R}$$

Let $\hat{u} = u^* - u$. From (7) we get

$$\partial_t \left( \frac{(u^*)^2}{2} \right) + \partial_x \mathcal{Q}(u^*) = u^* \mathcal{R}_{int,\theta*} \tag{32}$$

and from (31) we obtain

$$\partial_t \left( \frac{u^2}{2} \right) + \partial_x \mathcal{Q}(u) = 0 \tag{33}$$

Some calculation on (31) and (7) yields

$$\partial_t(u\hat{u}) + \partial_x(u(f(u^*) - f(u))) = [f(u^*) - f(u) - \hat{u}f'(u)]u_x + u\mathcal{R}_{int,\theta*} \tag{34}$$

Subtracting (34) and (33) from (32) we get

$$\partial_t S(u, u^*) + \partial_x H(u, u^*) = \hat{u}\mathcal{R}_{int,\theta*} + T_1 \tag{35}$$

with,

$$S(u, u^*) \coloneqq \frac{(u^*)^2}{2} - \frac{u^2}{2} - \hat{u}u = \frac{1}{2}\hat{u}^2,$$
$$H(u, u^*) \coloneqq \mathcal{Q}(u^*) - \mathcal{Q}(u) - u(f(u^*) - f(u)),$$
$$T_1 = -[f(u^*) - f(u) - f'(u)\hat{u}]u_x$$

As flux f is smooth, we can apply Taylor expansion[5] on $T_1$ and expand $f(u^*)$ at $u$,

$$T_1 = -\left[ \cancel{f(u)} + \cancel{f'(u)\hat{u}} + \frac{f''(u + \gamma(u^* - u))}{2}(u^* - u)^2 - \cancel{f(u)} - \cancel{f'(u)\hat{u}} \right] u_x$$
$$[\text{ where } \gamma \in (0, 1)]$$
$$= -\frac{1}{2} f''(u + \gamma(u^* - u)) \hat{u}^2 u_x$$
$$= -\frac{1}{2}\hat{u}^2 u_x$$

Hence it can be reasonably bounded by

$$|T_1| \leq \|u_x\|_{L^\infty} \hat{u}^2 \tag{36}$$

---

[5]with the Lagrange form of the remainder

where $C_{u_x}$ is given by $C_{u_x} = \|u_x\|_{L^\infty}$. Next, we integrate (35) over the domain (-1,1)

$$\int_{-1}^{1} \partial_t S(u, u^*) dx = -\int_{-1}^{1} \partial_x H(u, u^*) dx + \int_{-1}^{1} \hat{u} \mathcal{R}_{int,\theta^*} dx + \int_{-1}^{1} T_1 dx$$

$$\Rightarrow \frac{d}{dt} \int_{-1}^{1} \frac{\hat{u}^2(x,t)}{2} dx \leq H(u(-1,t), u^*(-1,t)) - H(u(1,t), u^*(1,t))$$

$$+ C_{u_x} \int_{-1}^{1} \hat{u}^2(x,t) dx + \int_{-1}^{1} \hat{u}(x,t) \mathcal{R}_{int,\theta^*}(x,t) dx$$

$$\Rightarrow \frac{d}{dt} \int_{-1}^{1} \hat{u}^2(x,t) dx \leq 2H(u(-1,t), u^*(-1,t)) - 2H(u(1,t), u^*(1,t))$$

$$+ 2C_{u_x} \int_{-1}^{1} \hat{u}^2(x,t) dx + \int_{-1}^{1} (\mathcal{R}_{int,\theta^*}^2(x,t) + \hat{u}^2(x,t)) dx$$

$$\Rightarrow \frac{d}{dt} \int_{-1}^{1} \hat{u}^2(x,t) dx \leq 2H(u(-1,t), u^*(-1,t)) - 2H(u(1,t), u^*(1,t))$$

$$+ C \int_{-1}^{1} \hat{u}^2(x,t) dx + \int_{-1}^{1} \mathcal{R}_{int,\theta^*}^2(x,t) dx \qquad (37)$$

where $C = 1 + 2C_{u_x}$. We can estimate $H(u(1,t), u^*(1,t))$ using (6)

$$H(u(1,t), u^*(1,t)) = \mathcal{Q}(u^*(1,t)) - \mathcal{Q}(u(1,t)) - u(1,t)(f(u^*(1,t)) - f(u(1,t)))$$

$$= \mathcal{Q}'(\gamma_1 \mathcal{R}_{sb,1,\theta^*}(t)) \mathcal{R}_{sb,1,\theta^*}(t) - \frac{u(1,t)}{2} [\hat{u}(1,t) [u^*(1,t) + u(1,t)]]$$

$$[ \text{ for some } \gamma_1 \in (0,1) \text{ by the mean-value theorem}]$$

$$= \gamma_1 f'(\gamma_1 \mathcal{R}_{sb,1,\theta^*}) \mathcal{R}_{sb,1,\theta^*}^2(t) - \frac{u(1,t)}{2} [\mathcal{R}_{sb,1,\theta^*} + 2u(1,t)] \mathcal{R}_{sb,1,\theta^*}$$

$$= \gamma_1 \left[ f'(\gamma_1 \mathcal{R}_{sb,1,\theta^*}) - \frac{u(1,t)}{2} \right] \mathcal{R}_{sb,1,\theta^*}^2(t) - u^2(1,t) \mathcal{R}_{sb,1,\theta^*}$$

$$\leq C_{2b} \mathcal{R}_{sb,1,\theta^*}^2(t) + u^2(1,t) |\mathcal{R}_{sb,1,\theta^*}|$$

$$\text{with } C_{2b} = C_{2b}(\|f'\|_\infty, \|u\|_{C^1([-1,1] \times [-1+\delta, \delta])})$$

Similarly we can estimate

$$H(u(-1,t), u^*(-1,t)) \leq C_{2b} \mathcal{R}_{sb,-1,\theta^*}^2(t) + u^2(-1,t) |\mathcal{R}_{sb,-1,\theta^*}|$$

Now, we can integrate (37) over the time interval $[-1+\delta, \bar{\delta}]$ for any $\bar{\delta} \in [-1+\delta, \delta]$ and use the above inequalities along with (8)

$$\int_{-1+\delta}^{\bar{\delta}} \frac{d}{dt} \int_{-1}^{1} \hat{u}^2(x,t) dx dt \leq \int_{-1+\delta}^{\bar{\delta}} (2H(u(-1,t), u^*(-1,t)) - 2H(u(1,t), u^*(1,t))) dt$$

$$+ \int_{-1+\delta}^{\bar{\delta}} C \int_{-1}^{1} \hat{u}^2(x,t) dx dt + \int_{-1+\delta}^{\bar{\delta}} \int_{-1}^{1} \mathcal{R}_{int,\theta^*}^2(x,t) dx dt$$

$$\Rightarrow \int_{-1}^{1} \hat{u}^2(x,\bar{\delta}) dx \leq \int_{-1}^{1} \hat{u}^2(x,-1+\delta) dx + 2C_{2b} \left[ \int_{-1+\delta}^{\delta} \mathcal{R}_{sb,-1,\theta^*}^2(t) dt + \int_{-1+\delta}^{\delta} \mathcal{R}_{sb,1,\theta^*}^2(t) dt \right]$$

$$+ 2 \left[ \int_{-1+\delta}^{\delta} u^2(-1,t) |\mathcal{R}_{sb,-1,\theta^*}| dt + \int_{-1+\delta}^{\delta} u^2(1,t) |\mathcal{R}_{sb,1,\theta^*}| dt \right]$$

$$+ C \int_{-1+\delta}^{\bar{\delta}} \int_{-1}^{1} \hat{u}^2(x,t) dx dt + \int_{-1+\delta}^{\delta} \int_{-1}^{1} \mathcal{R}_{int,\theta^*}^2(x,t) dx dt$$

$$\Rightarrow \int_{-1}^{1} \hat{u}^2(x,\bar{\delta})dx \le \int_{-1}^{1} \mathcal{R}_{tb,\theta^*}(x)dx + 2C_{2b}\left[\int_{-1+\delta}^{\delta} \mathcal{R}_{sb,-1,\theta^*}^2(t)dt + \int_{-1+\delta}^{\delta} \mathcal{R}_{sb,1,\theta^*}^2(t)dt\right]$$

$$+ 2C_{1b}\left[\int_{-1+\delta}^{\delta} |\mathcal{R}_{sb,-1,\theta^*}|dt + \int_{-1+\delta}^{\delta} |\mathcal{R}_{sb,1,\theta^*}|dt\right]$$

$$+ C\int_{-1+\delta}^{\bar{\delta}}\int_{-1}^{1} \hat{u}^2(x,t)dxdt + \int_{-1+\delta}^{\delta}\int_{-1}^{1} \mathcal{R}_{int,\theta^*}^2(x,t)dxdt$$

where $C_{1b} = C_{1b}(\|u(1,t)\|_{L^\infty})$

$$\le \int_{-1}^{1} \mathcal{R}_{tb,\theta^*}(x)dx + 2\bar{C}_{2b}\left[\int_{-1+\delta}^{\delta} \mathcal{R}_{sb,-1,\theta^*}^2(t)dt + \int_{-1+\delta}^{\delta} \mathcal{R}_{sb,1,\theta^*}^2(t)dt\right]$$

$$+ 2C_{1b}(\delta-(\delta-1))^{\frac{1}{2}}\left[\left(\int_{-1+\delta}^{\delta} \mathcal{R}_{sb,-1,\theta^*}^2 dt\right)^{\frac{1}{2}} + \left(\int_{-1+\delta}^{\delta} \mathcal{R}_{sb,1,\theta^*}^2 dt\right)^{\frac{1}{2}}\right]$$

$$+ C\int_{-1+\delta}^{\bar{\delta}}\int_{-1}^{1} \hat{u}^2(x,t)dxdt + \int_{-1+\delta}^{\delta}\int_{-1}^{1} \mathcal{R}_{int,\theta^*}^2(x,t)dxdt$$

by using Holder's inequality

$$\le C_T + C\int_{-1+\delta}^{\bar{\delta}}\int_{-1}^{1} \hat{u}^2(x,t)dxdt$$

where $C_T = \int_{-1}^{1} \mathcal{R}_{tb,\theta^*}(x)\,\mathrm{d}x + 2C_{2b}\left[\int_{-1+\delta}^{\delta} \mathcal{R}_{sb,-1,\theta^*}^2(t)\,\mathrm{d}t + \int_{-1+\delta}^{\delta} \mathcal{R}_{sb,1,\theta^*}^2(t)\,\mathrm{d}t\right]$

$$+ \int_{-1+\delta}^{\delta}\int_{-1}^{1} \mathcal{R}_{int,\theta^*}^2(x,t)\,\mathrm{d}x\,\mathrm{d}t + 2C_{1b}\left[\left(\int_{-1+\delta}^{\delta} \mathcal{R}_{sb,-1,\theta^*}^2\,\mathrm{d}t\right)^{\frac{1}{2}} + \left(\int_{-1+\delta}^{\delta} \mathcal{R}_{sb,1,\theta^*}^2\,\mathrm{d}t\right)^{\frac{1}{2}}\right] \quad (38)$$

Using integral form of Grönwall's inequality on (38)

$$\int_{-1}^{1} \hat{u}^2(x,\bar{\delta})dx \le C_T + \int_{-1+\delta}^{\bar{\delta}} C_T C e^{\int_t^\delta C ds}dt \le \left[1 + \int_{-1+\delta}^{\bar{\delta}} C e^C dt\right]C_T \quad (39)$$

Integrating (39) over $\bar{\delta}$ together with the definition of generalization error (10) we get

$$\int_{-1+\delta}^{\delta}\int_{-1}^{1} \hat{u}^2(x,\bar{\delta})dxd\bar{\delta} \le C_T \int_{-1+\delta}^{\delta}\left[1 + \int_{-1+\delta}^{\bar{\delta}} C e^C dt\right]d\bar{\delta}$$

$$\mathcal{E}_G^2 \le \left[1 + Ce^C\right]C_T \quad (40)$$

$\square$

## A.4 MAKING THE DATA DEPENDENCE EXPLICIT IN THE BOUNDS FOR $1+1$ BURGERS' PDE

### A.4.1 QUADRATURE RULE

Let' say we have a mapping $g: \mathbb{D} \to \mathbb{R}^m$ such that $g \in Z^* \subset L^p(\mathbb{D}, \mathbb{R}^m)$ and $\mathbb{D} \subset \mathbb{R}^{\bar{d}}$. Let's say we have an integral that we want to approximate

$$\bar{g} := \int_{\mathbb{D}} g(y)\,dy \quad (41)$$

where $dy$ denotes the $\bar{d}$-dimensional Lebesgue measure. To approximate this integral by the quadrature rule we need (i) the quadrature points $y_i \in \mathbb{D}$ for $1 \le i \le N$ for some $N \in \mathbb{N}$ and (ii) weights $w_i$ with $w_i \in \mathbb{R}_+$. Then we can approximate (41) by the quadrature

$$\bar{g}_N := \sum_{i=1}^{N} w_i g(y_i) \quad (42)$$

Then the error of this approximation is bounded by

$$|\bar{g} - \bar{g}_N| \le C_{quad}(\|g\|_{Z^*}, \bar{d})\, N^{-\alpha}, \text{ for some } \alpha > 0 \quad (43)$$

These quadrature weights, quadrature points and $\alpha$ vary with $\bar{d}$'s range.

### A.4.2 Applying Quadrature rule on Theorem 3.2

The loss function can then be written as

$$\mathcal{L}(\theta) = \mathcal{E}_T^2 := \underbrace{\frac{1}{N_{tb}} \sum_{n=1}^{N_{tb}} w_n^{tb} |\mathcal{R}_{tb,\theta^*}(x_n)|^2}_{(\mathcal{E}_T^{tb})^2} + \underbrace{\frac{1}{N_{sb}} \sum_{n=1}^{N_{sb}} w_n^{sb} |\mathcal{R}_{sb,-1,\theta^*}(t_{n,\delta})|^2}_{(\mathcal{E}_T^{sb,-1})^2}$$

$$+ \underbrace{\frac{1}{N_{sb}} \sum_{n=1}^{N_{sb}} w_n^{sb} |\mathcal{R}_{sb,1,\theta^*}(t_{n,\delta})|^2}_{(\mathcal{E}_T^{sb,1})^2} + \underbrace{\frac{\lambda}{N_{int}} \sum_{n=1}^{N_{int}} w_n^{int} |\mathcal{R}_{int,\theta^*}(x_n,t_{n,\delta})|^2}_{(\mathcal{E}_T^{int})^2} \tag{44}$$

In our experiments we choose all $w$-s to be equal to $1$ and train our model on that

**Theorem A.2.** *Let $u \in C^k((-1+\delta,\delta) \times (-1,1))$ be the unique solution of the viscous scalar conservation law for any $k \geq 1$. Let $u^* = u_{\theta^*}$ be the PINN, then the generalization error (10) is bounded by*

$$\mathcal{E}_G^2(u^*) \leq \left(1 + Ce^C\right) \left[ \sum_{n=1}^{N_{tb}} w_n^{tb} |\mathcal{R}_{tb,\theta^*}(x_n)|^2 + \sum_{n=1}^{N_{int}} w_n^{int} |\mathcal{R}_{int,\theta^*}(x_n,t_{n,\delta})|^2 \right.$$

$$+ 2C_{2b} \left( \sum_{n=1}^{N_{sb}} w_n^{sb} |\mathcal{R}_{sb,-1,\theta^*}(t_{n,\delta})|^2 + \sum_{n=1}^{N_{sb}} w_n^{sb} |\mathcal{R}_{sb,1,\theta^*}(t_{n,\delta})|^2 \right) + 2C_{1b}\left(\mathcal{E}_T^{sb,-1} + \mathcal{E}_T^{sb,1}\right)$$

$$+ \frac{C_{quad}^{tb}}{N_{tb}^{\alpha_{tb}}} + \frac{C_{quad}^{int}}{N_{int}^{\alpha_{int}}} + 2C_{2b}\left(\frac{\left(C_{quad}^{sb,-1} + C_{quad}^{sb,1}\right)}{N_{sb}^{\alpha_{sb}}}\right) + 2C_{1b}\left(\frac{\left(C_{quad}^{sb,-1} + C_{quad}^{sb,1}\right)}{N_{sb}^{\frac{\alpha_{sb}}{2}}}\right) \right] \tag{45}$$

*where $C = 1 + 2C_{u_x}$, with*

$$C_{u_x} = \|u_x\|_{L^\infty} = \left\|\frac{1}{t-1}\right\|_{L^\infty([-1+\delta,\delta])}$$

$$C_{1b} = \|u(1,t)\|_{L^\infty}^2 = \left\|\frac{1}{1-t}\right\|_{L^\infty([-1+\delta,\delta])}^2$$

$$C_{2b} = \|u_{\theta^*}(1,t)\|_{L^\infty([-1+\delta,\delta])} + \frac{3}{2}\left\|\frac{1}{t-1}\right\|_{L^\infty([-1+\delta,\delta])} \tag{46}$$

*and $C_{quad}^{tb} = C_{quad}^{tb}\left(\|\mathcal{R}_{tb,\theta^*}^2\|_{C_k}\right)$, $C_{quad}^{int} = C_{quad}^{int}\left(\|\mathcal{R}_{int,\theta^*}^2\|_C^{k-2}\right)$, $C_{quad}^{sb,-1} = C_{quad}^{sb,-1}\left(\|\mathcal{R}_{sb,-1,\theta^*}\|_{C^k}\right)$, $C_{quad}^{sb,1} = C_{quad}^{sb,1}\left(\|\mathcal{R}_{sb,1,\theta^*}^2\|_{C^k}\right)$ are constants of the quadrature bound.*

*Proof.* In equation (40) within the proof of Theorem 3.2 we see that

$$\int_{-1+\delta}^{\delta} \int_{-1}^{1} \hat{u}^2(x,\bar{\delta}) dx d\bar{\delta} \leq C_T \int_{-1+\delta}^{\delta} \left[1 + \int_{-1+\delta}^{\bar{\delta}} Ce^C dt\right] d\bar{\delta}$$

$$\mathcal{E}_G^2 \leq \left[1 + Ce^C\right] C_T$$

where

$$C_T = \underbrace{\int_{-1}^{1} \mathcal{R}_{tb,\theta^*}(x) dx}_{1} + \underbrace{2C_{2b}\left[\int_{-1+\delta}^{\delta} \mathcal{R}_{sb,-1,\theta^*}^2(t) dt + \int_{-1+\delta}^{\delta} \mathcal{R}_{sb,1,\theta^*}^2(t) dt\right]}_{3}$$

$$+ \underbrace{\int_{-1+\delta}^{\delta} \int_{-1}^{1} \mathcal{R}_{int,\theta^*}^2(x,t) dx dt}_{2} + \underbrace{2C_{1b}\left[\left(\int_{-1+\delta}^{\delta} \mathcal{R}_{sb,-1,\theta^*}^2 dt\right)^{\frac{1}{2}} + \left(\int_{-1+\delta}^{\delta} \mathcal{R}_{sb,1,\theta^*}^2 dt\right)^{\frac{1}{2}}\right]}_{4} \tag{47}$$

Applying quadrature bounds on (47) we get,

$$
C_T \leq \underbrace{\sum_{n=1}^{N_{tb}} w_n^{tb} |\mathcal{R}_{tb,\theta^*}(x_n)|^2 + C_{quad}^{tb}(\|\mathcal{R}_{tb,\theta^*}\|_{C^k})N_{tb}^{-\alpha_{tb}}}_{1}
$$

$$
+ \underbrace{\sum_{n=1}^{N_{int}} w_n^{int} |\mathcal{R}_{int,\theta^*}(x_n, t_{n,\delta})|^2 + C_{quad}^{int}(\|\mathcal{R}_{int,\theta^*}\|_{C^{k-2}})N_{int}^{-\alpha_{int}}}_{2}
$$

$$
+ 2C_{2b}\Bigg[ \underbrace{\sum_{n=1}^{N_{sb}} w_n^{sb} |\mathcal{R}_{sb,-1,\theta^*}(t_{n,\delta})|^2 + \sum_{n=1}^{N_{sb}} w_n^{sb} |\mathcal{R}_{sb,1,\theta^*}(t_{n,\delta})|^2}_{3}
$$

$$
+ \underbrace{\left( C_{quad}^{sb}(\|\mathcal{R}_{sb,-1,\theta^*}\|_{C^k}) + C_{quad}^{sb}(\|\mathcal{R}_{sb,1,\theta^*}\|_{C^k}) \right) N_{sb}^{\alpha_{sb}}}_{3} \Bigg]
$$

$$
+ 2C_{1b}\Bigg[ \underbrace{\left( \sum_{n=1}^{N_{sb}} w_n^{sb} |\mathcal{R}_{sb,-1,\theta^*}(t_{n,\delta})|^2 \right)^{\frac{1}{2}} + \left( \sum_{n=1}^{N_{sb}} w_n^{sb} |\mathcal{R}_{sb,1,\theta^*}(t_{n,\delta})|^2 \right)^{\frac{1}{2}}}_{4}
$$

$$
+ \underbrace{\left( C_{quad}^{sb}(\|\mathcal{R}_{sb,-1,\theta^*}\|_{C^k}) + C_{quad}^{sb}(\|\mathcal{R}_{sb,1,\theta^*}\|_{C^k}) \right)^{\frac{1}{2}} N_{sb}^{\frac{\alpha_{sb}}{2}}}_{4} \Bigg] \tag{48}
$$

Replacing the sums of residuals by training error we get

$$
\mathcal{E}_G^2 \leq \left(1 + Ce^C\right)\Big[ \left(\mathcal{E}_T^{tb}\right)^2 + \left(\mathcal{E}_T^{int}\right)^2 + 2C_{2b}\left( \left(\mathcal{E}_T^{sb,-1}\right)^2 + \left(\mathcal{E}_T^{sb,1}\right)^2 \right) + 2C_{1b}\left( \mathcal{E}_T^{sb,-1} + \mathcal{E}_T^{sb,1} \right) \Big]
$$

$$
+ \left(1 + Ce^C\right)\Big[ C_{quad}^{tb}N_{tb}^{-\alpha_{tb}} + C_{quad}^{int}N_{int}^{-\alpha_{int}} + 2C_{2b}\left( \left(C_{quad}^{sb,-1} + C_{quad}^{sb,1}\right)N_{sb}^{-\alpha_{sb}} \right)
$$

$$
+ 2C_{1b}\left( \left(C_{quad}^{sb,-1} + C_{quad}^{sb,1}\right)N_{sb}^{\frac{-\alpha_{sb}}{2}} \right) \Big] \tag{49}
$$

$\square$

## B  PLOTTING THE BEHAVIOUR OF RHS OF EQUATION 11 WITH VARYING WIDTH

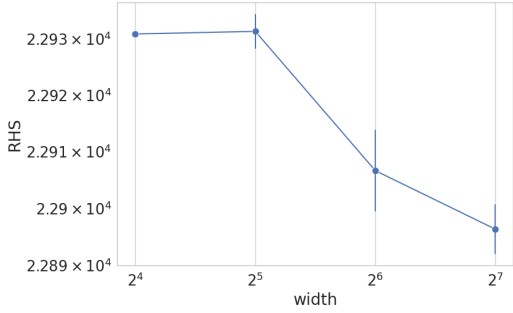

Figure 3: This plot tracks the RHS of equation (11) in Theorem 3.2 for training a depth 2 net at different widths towards solving equation 6 at $\delta = \frac{1}{2}$

## C   PLOTTING THE NEURALLY DERIVED SOLUTION FOR EQUATION 6 (LEFT) AND THE TRUE SOLUTION (RIGHT)

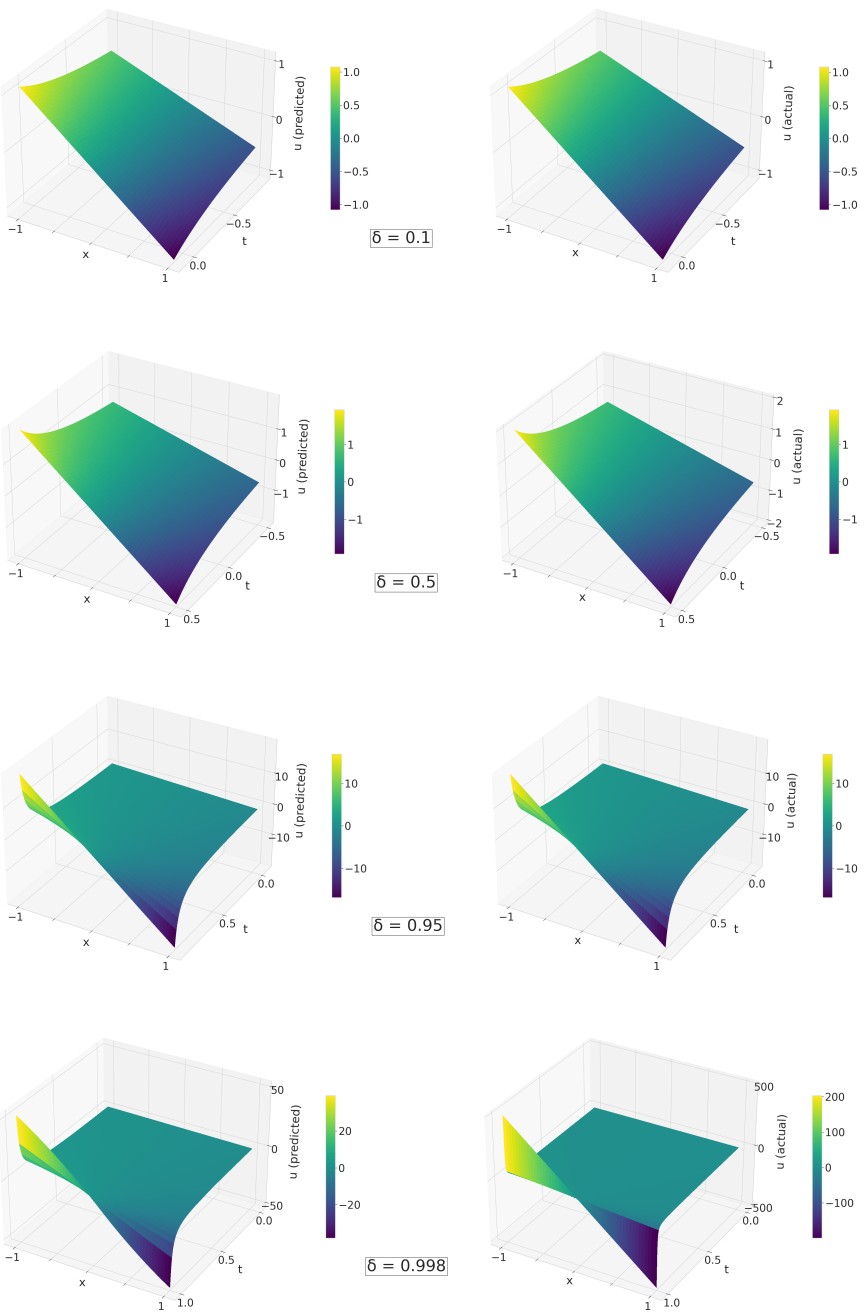

Figure 4: A demonstration of the visual resemblance between the neurally derived solution for equation 6 (left) and the true solution (right) at different values of the $\delta$ parameter getting close to the PDE with blow-up at $\delta = 1$. A PINN with a width of 300 and a depth of 6 was trained to generate the plots on the left.

# D A STUDY OF THE APPROXIMATE INVARIANCE OF THE TRAINING TIME FOR THE 1D BURGERS' PDE VS PROXIMITY TO SINGULARITY

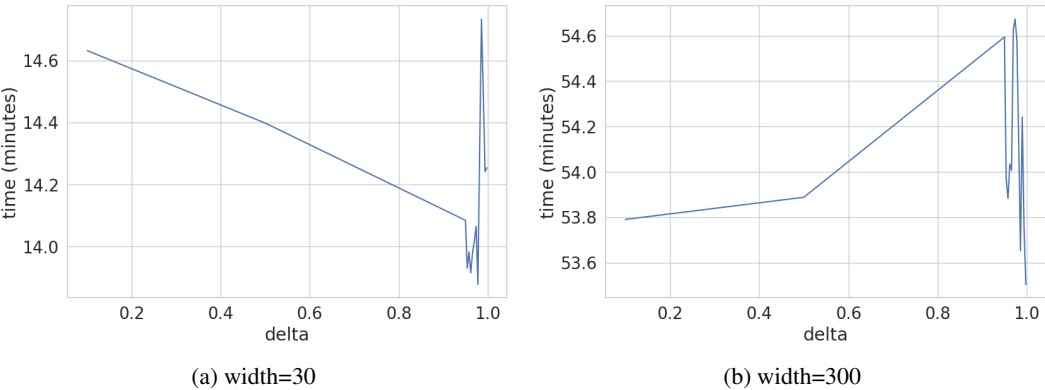

(a) width=30                            (b) width=300

Figure 5: These plots show that the time taken to train a PINN on equation 6 barely changes for different values of $\delta$ - a measure of proximity to blow-up and that this holds at two widely separated widths of the net.

# E INAPPLICABILITY OF CLASSICAL NUMERICAL ANALYSIS ERROR BOUNDS TO PINN EXPERIMENTS

To the best of our knowledge existing results in numerical analysis cannot be deployed to understand PINN training - as is the target here. Specifically for the $0$ viscosity Burgers' PDE one can see that in works like Johnson & Szepessy (1987), the theory does not seem to be give a bound on the distance from the true solution of the solution found by the finite element method.

More generally, in works such as Corollary 3.5 in Tadmor (1991) the authors consider a weak solution of the $\varepsilon$-viscosity regularized Burgers' PDE and derive bounds on the local $L^p$-distance between the weak solution and the true solution at zero viscosity. There is no obvious way to apply these bounds for a PINN solution since the trained net has no guarantee to be satisfying the conditions required of the surrogate here.

For results like Theorem 2.1 in Nessyahu & Tadmor (1992), we observe that these too don't have an obvious way for the bounds to be applied for PINN experiments because they need stringent conditions (like satisfying the conservativeness property) to be true for the approximant, for the bounds to apply and there is no natural way to know if the neural surrogate satisfies these conditions. Also both the above cited classical bounds are not tailored to any compact domain and hence there is no boundary condition error that is getting tracked there as in our Theorem 3.2.

