# OpenReview forum: "Investigating the Ability of PINNs To Solve Burgers' PDE Near Finite-Time BlowUp"
_ICLR.cc/2024/Conference — Submitted to ICLR 2024_

### Official Review · Reviewer_aEZi · 2023-10-28

**Soundness:** 3 good
**Presentation:** 3 good
**Contribution:** 3 good
**Rating:** 6
**Confidence:** 4

**Summary:**

This paper studies the generalization error of PINN for Burgers' equation. The theoretical framework is informative of the empirical evaluations.

**Strengths:**

This paper innovatively studies the PINN generalization error of the Burgers equation.
Empirical evaluation validates the effectiveness of the theoretical framework.
The bound does not depend heavily on the neural network.
The solution of the Burgers equation is stiff, which hinders PINN from learning this part of the mutation. Therefore, the topic studied in the paper is important.

**Weaknesses:**

Although I recognize the theoretical contribution of this paper, the actual PINN experiment deviates from the theory to a certain extent.
Because the solution to the Burgers equation is very stiff, many PINN variants have been proposed to solve these problems, such as self-adaptive weight PINN, adaptive sampling, or adversarial training. Their core points are to focus the optimization of PINN on these stiff areas with relatively large losses to fit the stiff area of the Burgers equation well.
Since the theory of this paper is mainly based on PINN's L2 loss to bound the final generalization error. Therefore, I suspect that the conclusions of this paper cannot fit well with these PINN variants, such as self-adaptive weight PINN, adaptive sampling, or adversarial training, because the loss function they use is no longer L2 loss. In other words, the most popular method to solve Burger is adaptive loss. Can the author's theoretical framework be applicable to these variants?

**Questions:**

See the weakness part.

---

> ### Author Response · Authors · 2023-11-16
> **Our bounds can be invoked on the approximate solution obtained from any kind of a PINN loss.**
>
> We thank you for your kind comments about our work.
> In the following, we address the key concern you have raised.
>
> > Can the author's theoretical framework be applicable to PINN variants, such as self-adaptive weight PINN, adaptive sampling, or adversarial training?
>
> Although our bounds in the theorems are computing errors in the $L^2$ norm, the proof can be applied to any surrogate solution found irrespective of the method used to obtain it (like the choice of norms in the loss function used) - in other words, our bounds will hold as long as the initial and boundary conditions are kept similar to ours for any method solving the Burgers' PDE and returning a function as a candidate solution.
>
> For eg. in self-adaptive weighting (https://arxiv.org/abs/2009.04544) they modify the loss function by adding specific weights to each residual and these weights are updated in each time step. In adaptive sampling (https://arxiv.org/abs/2210.00279), they choose a set of points in each step of training on the basis of ``failure probability'' at a few points in space. In both of these cases, the modifications do not affect the applicability of our bound since that only depends on being able to estimate the required boundary, initial conditions and bulk error computing integrals for the surrogate at the end of training - howsoever obtained.

---

> > ### Author Response · Authors · 2023-11-21
> > **Thanks!**
> >
> > Given the rebuttal process is coming to an end in a couple of days, we would be grateful to know if the reviewer has found our answers to be satisfactorily answering all their questions. Kindly feel free to let us know if there are additional queries and we would be happy to strive to clarify them.

---

### Official Review · Reviewer_ndTQ · 2023-10-30

**Soundness:** 3 good
**Presentation:** 3 good
**Contribution:** 3 good
**Rating:** 6
**Confidence:** 4

**Summary:**

The paper starts by highlighting a gap: current general rules for using neural networks to solve PDEs don't exist if the PDE has a known explosive solution. This pushes the authors to explore how PINNs tackle the Burgers' PDE, especially when it's close to exploding.

First, the authors explain PINNs. These are neural networks trained to follow the rules of a physical system, including its boundary and starting points. They then test how well PINNs handle the challenging parts of Burgers' PDE and compare this to older, standard methods.

After that, they work out general rules for errors in the Burgers' PDE. These rules estimate how much the neural network might get wrong on new data. The authors find a link between these rules and the solution the neural network comes up with. They suggest these rules can help shape how the neural network is built.

The paper then talks about the balance between getting the answer quickly and getting it right in PINNs. The authors suggest a new training method for PINNs that finds a good middle ground. They test this on Burgers' PDE and find it gives good answers much faster than older methods.

To sum up, this paper adds a lot to the world of using neural networks to solve PDEs. It shows how PINNs can handle tough PDE situations, gives rules for estimating errors, and introduces a faster training method. All these can shape how future neural networks are designed for this job, leading to quicker, more accurate results.

**Strengths:**

Fresh Perspective: The paper delves into how Physics Informed Neural Networks (PINNs) handle particular solutions in PDEs, a topic not widely tackled before. Additionally, the authors outline error estimation rules for the Burgers' PDE when using neural networks, marking a pioneering step in neural network-based PDE solutions.

Thoroughness: The study dives deep into PINNs' stability, providing a well-rounded theoretical perspective. The authors craft error rules for the Burgers' PDE rooted in robust mathematical studies, bolstering the case for using PINNs to solve PDEs.

Practical Tests: The team showcases how PINNs can manage the Burgers' PDE, especially when it's on the brink of a complex issue, and stack these results against established methods. They also suggest and test a fresh PINN training technique that strikes a balance between speed and precision. These hands-on results further confirm the potential of PINNs in this domain.

Clear Writing: The paper is neatly composed and straightforward. With lucid explanations and detailed accounts of their methods and findings, it caters to a broad audience, even those just venturing into neural network-based PDE solutions.

**Weaknesses:**

One limitation of this paper is its narrow focus on addressing the Burgers' PDE near a specific complex scenario. Although this is a significant topic, it might not cover the spectrum of PDEs used in real-world situations. This could limit how much the findings in this paper can be applied to other PDEs.

Furthermore, the study works under the assumption that we always know the main equations driving the physical system. However, in real situations, these equations might be unknown or hard to pinpoint. This could reduce the range of situations where PINNs can be effectively used for solving PDEs.

Lastly, the paper could have delved deeper into comparing its method with other leading neural network solutions for PDEs. While there's a comparison with classic numerical methods, a broader analysis including other neural network strategies would give readers a fuller understanding of where this method stands in the landscape of PDE-solving techniques.

**Questions:**

What is the trade-off between accuracy and speed of inference in PINNs?

How do PINNs detect finite-time blow-ups in PDEs?

What are the generalization bounds for Burgers' PDE and how are they correlated to the neurally found surrogate solution?

---

> ### Author Response · Authors · 2023-11-16
> **Added new experiment details and literature reviews in Appendices C, D and E**
>
> We thank the reviewer for their detailed analysis of our paper and the kind comments.
>
> Firstly, we would like to point out that in the newly added Appendix C we have now given a side-by-side comparison between the true PDE solution and the neurally found approximate PDE solution for a sequence setups with 1D Burgers' PDE whose true solutions are increasingly getting close to being singular.
>
> Secondly, please see Appendix D where we have demonstrated how the run-time to solve 1D Burgers' PDE by the PINN method is barely affected by the proximity to the singularity of the true solution.
>
> Thirdly, kindly note the new Appendix E where we have reviewed that to the best of our knowledge existing results in the classical numerical analysis literature don't seem to be applicable to analyze PINN experiments with Burgers' PDE.
>
> In the following we shall address the key concerns you have raised.
>
> >it might not cover the spectrum of PDEs used in real-world situations
>
> To the best of our knowledge there is hardly any universal principle for how finite-time singularities appear in different PDEs. In our introduction, we have reviewed the evidence for such singularities in various famous models like the Frank-Kamenetskii PDE, 2D Boussinesq PDE, and the 3D Euler PDE. Each has its own peculiarities and mechanisms of blowing-up and we believe that each will require its own new analysis.
>
> We hope that our work could be a starting point for various such analyses in the future.
>
> >The paper could have delved deeper into comparing its method with other leading neural network solutions for PDEs.
>
> The theoretical framework proposed by us can be invoked on *any* surrogate solution found for the specified Burgers' PDE. The *applicability of our theoretical bounds does not depend on any specific method used for solving the PDEs* as long as the chosen numerical method gives a function approximation to the true PDE solution - as PINNs do, which is the primary method we focus on.
>
>
> >the study works under the assumption that we always know the main equations driving the physical system.
>
> In this work we focus on PINNs, which is a method of using deep-learning to solve PDEs in an unsupervised way - and by definition, this method is applicable only when the exact PDE is known. We posit that this is a common use case in the real world - and why PINNs are getting increasingly deployed for various uses.
>
> In cases where the underlying PDE is unknown but some input-output function samples (like samples of solution for different initial conditions) are available, operator methods such as DeepONets can be used. Formulating the problem of stability of operator learning in the face of finite-time blow-ups is an entirely different question (about learning in infinite dimensions) than what we focus on.
>
> >What are the generalization bounds for Burgers' PDE and how are they correlated to the neurally found surrogate solution?
>
> In usual ML parlance when we want to prove bounds on the generalization error for a setup, we are looking to bound the difference between the population risk and the empirical risk of a loss function. As we have pointed out in the review given in our Section 2, such bounds for PDE solving losses is available only for special kinds of linear PDE. Burgers' PDE being non-linear is not in the ambit of such existing proofs - and that is one among the many motivations for our work.
>
> Towards such a goal we posit that it could be an interesting direction of future exploration to try to understand the Rademacher complexity of the loss class considered here.

---

> > ### Author Response · Authors · 2023-11-21
> > **Thanks!**
> >
> > Given the rebuttal process is coming to an end in a couple of days, we would be grateful to know if the reviewer has found our answers to be satisfactorily answering all their questions. Kindly feel free to let us know if there are additional queries and we would be happy to strive to clarify them.

---

### Official Review · Reviewer_7pxF · 2023-10-31

**Soundness:** 3 good
**Presentation:** 3 good
**Contribution:** 2 fair
**Rating:** 5
**Confidence:** 2

**Summary:**

In this paper, the approximation ability of PINNs for the inviscid Burgers equation is theoretically estimated. This equation is known to have so-called blow-up solutions, which are solutions that diverge to infinity in finite time. In this paper, whether PINNs can find such a solution is investigated theoretically. Specifically, two theorems are presented in this paper; the former theorem gives an error estimate for the multi-dimensional Burgers equation, and the latter theorem gives an improved result for the 1-dimensional equation.

**Strengths:**

This is just my impression but theoretical error analysis of numerical methods for computing blow-up solutions is a difficult problem. Even for classical numerical methods, such as the finite difference method and the finite volume method, there are not so many papers on this topic. A strength of this paper is that the authors tackle such a challenging problem, and certain results are in fact given.

**Weaknesses:**

I suppose that there are a few weaknesses in this paper.
1) I believe that inequalities estimating numerical errors should show that the error bound converges to zero in some sense. If I understand the result correctly, the error bound in the first theorem does not converge to zero because $C_1$ and $C_2$ include the terms given by the norm of the solutions.  So, the inequality (5) does not appear to make sense as an error analysis.

2) As for the results of the numerical experiments, although it is interesting that certain correlations between RHS and LHS of the inequalities are observed, the magnitudes of them are very different. So, I am not sure whether these results are meaningful or not.

3) Perhaps this is not a weakness, but honestly, it is difficult for me to assess the value of this paper in the ML community. Although the analysis shown in this paper may be an important first step in this direction, I am not sure whether the results of this paper meet the criteria of a top ML conference. My concern is that, in my impression, papers on error analysis of classical numerical methods (e.g., the finite difference method) for the Burgers equation seem unlikely to be accepted by top journals of numerical analysis because the Burgers equation is the simplest partial differential equation with blow-up solutions.

**Questions:**

My biggest concern is the first one of the above weaknesses. Does the error bound (5) converge to zero in certain situations?

---

> ### Author Response · Authors · 2023-11-15
> **Added new experiment details and literature reviews in Appendices C, D and E**
>
> We thank the reviewer for their kind reading of our work and for appreciating the challenge that we have tried to solve.
>
> Firstly, we would like to draw your attention to the newly added Appendix E in the draft where we have reviewed multiple classical results in numerical analysis for Burgers' PDE to establish that to the best of our knowledge even for this smallest non-trivial case of finite-time blow-up, classical error bounds do not exist which can be invoked on PINN experiments with Burgers' PDE.
>
> Thus, our bounds and experiments in this work can be seen to be a first-of-its-kind result to address a challenging interface between deep-learning and non-linear PDEs.
>
> >Does the error bound (5) converge to zero in certain situations?
>
> Thanks for raising this interesting question! As you would note we have specified this exact question as an open question in our conclusion in the language of "stability" - which we have briefly defined in the footnote on page 6.
>
> For PDEs for which the ML method of solving it has a "stability" guarantee in this precise sense are the very cases where we can get a data-driven sufficient condition for the error of the approximation to be $0$.
>
> As we have pointed out in our discussion below Theorem 3.2, that this property of stability is not immediate for many of the ML losses used for PDE solving and one of our key observations (in Theorem 3.2) is that for Burger's PDE in one dimensions such a stable bound is possible despite allowing for the difficult solution of a finite-time blow-up.
>
> >Although it is interesting that certain correlations between RHS and LHS of the inequalities are observed, the magnitudes of them are very different. So, I am not sure whether these results are meaningful or not.
>
> As we have emphasized in the literature review done at the beginning of Section $4$, getting non-vacuous generalization bounds for deep-learning is an extremely open problem. The objective of our research here is not to try to solve this question which is open for even much more basic setups with neural nets.
>
> The crux of our work - in particular our experiments - is to establish that the theoretical bounds we prove are not only applicable to this involved setup of nets solving PDEs but that they also evaluate to values that are highly correlated to the functional distance of the found surrogate from the true PDE solution.
>
> We posit that it is somewhat of a surprise that this correlation between the bound and the truth is strong (approaching nearly $1$ at times) despite the vacuity of the bound and despite the proximity to singularities - which is a unique feature of our PDE under consideration.

---

> > ### Author Response · Authors · 2023-11-21
> > **Thanks!**
> >
> > Given the rebuttal process is coming to an end in a couple of days, we would be grateful to know if the reviewer has found our answers to be satisfactorily answering all their questions. Kindly feel free to let us know if there are additional queries and we would be happy to strive to clarify them.

---

> > > ### Comment · Reviewer_7pxF · 2023-11-22
> > >
> > > Many thanks for your reply. I am afraid to say, but, although my impression of this paper is not so bad, my biggest concern in Questions has not been fully addressed. I agree with the authors that the bounds provided in this paper contribute to the stability analysis of PINNs for the blow-up solutions of the Burgers equation and this paper proceeds an important step in this direction; however, I believe that "generalization bounds" should converge to zero under certain assumptions. I suppose that this paper would be more readable and of greater value if it could be rewritten as a paper on stability analysis.

---

> ### Author Response · Authors · 2023-11-22
> **Our use of the term "generalization error" is consistent with foundational papers in PINN theory**
>
> Thanks for your comments. In usual ML, the word "generalization bound" typically refers to a bound on the gap between the population risk and the empirical risk and this can often be shown to have easy sufficient conditions of approaching $0$ by methods such as Rademacher complexity.
>
> *But in our theorems here, the LHS is not this notion of generalization gap* but a measure of the distance between the surrogate function and the true PDE solution. In the context of PINNs this quantity has been called as the "generalization error" right from the first theory papers in PINNs like the canonical result from 2022, Theorem 2.6 here,  https://doi.org/10.1093/imanum/drab093  - which laid the foundations of the subject. Please note that the authors there too called this gap from the true PDE solution as "generalization error"
>
> But kindly note that the bounds like in the foundational papers use an unconventional notion of training error (which carefully weights each collacation point) which is not implemented in PINN experiments these days but our bounds don't have such a condition - and thus we take a step forward compared to them and build insights closer to experimental setups.
>
> So, we posit that our use of the word generalization error for this quantity is consistent with the foundational papers in the theory of PINNs while we take a step forward in many other ways. Maybe we kindly request the reviewer to take cognizance of this fact.

---

### Official Review · Reviewer_QLQE · 2023-10-31

**Soundness:** 2 fair
**Presentation:** 2 fair
**Contribution:** 2 fair
**Rating:** 3
**Confidence:** 4

**Summary:**

This work derives what the authors call a generalization bound for the PINN-based solution of Burgers' equation near the formation of singularities. They show empirically that their bound, while vacuous, is surprisingly correlated with the error vs the true solution.

**Strengths:**

Introducing more analytical techniques to the study of PINN training is a worthwhile cause. I also appreciate the author's openness to admit the vacuousness of their bound and investigating the empirical correlation of their bound with the right hand side.

**Weaknesses:**

Listed in decreasing order of gravity

1. The bound derived by the authors depends on $L^\infty$ norm of the gradient. As the equation approaches blow-up, this quantity approaches infinity. The bound thus does not provide meaningful information in the vicinity of the blowup, which is undercutting the main claimed contribution.

2. The claim by the authors
>Most importantly, Theorem 3.2 shows that despite the setting here being of proximity to finite-time
blow-up, the naturally motivated PINN risk in this case 3
is “(L2, L2, L2, L2)-stable”4 in the precise sense as defined in Wang et al. (2022a). This stability property being true implies that if the PINN
risk of the solution obtained is measured to be O(ϵ) then it would directly imply that the L2-risk
with respect to the true solution (10) is also O(ϵ). And this would be determinable without having
to know the true solution at test time.

is misleading. If the exact solution is unknown, neither is the $L^\infty$ value of its gradient at a given time, preventing the bounding of the error vs the true solution.

3. I do find the expression "generalization bound" for Theorem 3.1 somewhat misleading. These type of stability estimates (of the operator mapping right hand side and initial condition to the solution) are standard tools in the theory of partial differential equations, making this seem more like a rebranding. It would strengthen the paper if the authors would discuss related results in the PDE literature.

4. The literature review on operator learning approaches misses the works on both neural operators and BCR-NET (the latter predates both neural operators and DeepONet).

5. The referral to the works on the euler singularity of Wang should make more clear the differences between the two works. To my understanding, the work of Wang et al uses a rescaled coordinate system and therefore does not actually solve a PDE with singular solution. The blow-up studied by this community is also specific to incompressible problems as the blowup of the compressible Euler equation (of which the Burgers equation is the zero sound speed limit) arises from a different phenomenon.

**Questions:**

I suggest the authors directly respond to my criticism in the last paragraph. I would gladly reconsider my recommendation if it turns out that I overlooked something.

---

> ### Author Response · Authors · 2023-11-15
> **Added new experiment details and literature reviews in Appendices C, D and E**
>
> We thank the reviewer for their critical reading of our work.
>
> May we request you to reconsider your scores in light of the explanations that we provide below for the $5$ weaknesses you have pointed out.
>
> > As the equation approaches blow-up, the $L^\infty$-norm of the gradient approaches infinity. The bound thus does not provide meaningful information in the vicinity of the blowup.
>
> We suggest reading our bounds not at the blow-up point but in its vicinity i.e when $\delta \in [0,1)$ for say Theorem $3.2$. At any such $\delta$ the bound is finite and that's when we do our experiments. The meaning we extract from these bounds is as displayed in Figures $1$ and $2$ where we show that despite the bounds being computed arbitrarily close to a blow-up they maintain correlation with the distance of the neurally found surrogate from the true solution.
>
> To the best of our knowledge, except this work, currently there are no other theorems available off-the-shelf which can be used to get any bound at all on the population risk of a neurally found surrogate solving for Burgers' PDE near a finite-time blow-up.
>
> We would like our work to be seen as a critical first step towards understanding the training of neural nets to solve PDEs near a blow-up.
>
> >[The $(L_2,L_2,L_2,L_2)$-stability claim] is misleading
>
> We would like to draw your attention to how stability has been defined in the footnote of page $6$. This definition is inspired from an earlier work, https://doi.org/10.48550/arXiv.2206.02016. As you would note that this definition is about an order estimate - that if the population risk of the PINN loss is ${\cal O}(\epsilon)$ then it should be a sufficient condition to ensure that the distance of the surrogate from the true PDE solution is also ${\cal O}(\epsilon)$.
>
> We note that in general for arbitrary PDE, it is not guaranteed (and may even be provably impossible) for the PINN loss to have this property of stability.
>
> And this is the special phenomenon that happens in our Theorem $3.2$.
>
> Thus even when the true solution is unknown, a PINN loss could be proven to be stable if the bound on the true risk can be shown to depend on the true solution such that the dependencies can be absorbed into the constants kept implicit in the ${\cal O}$ estimate.
>
> >"generalization error" term for Theorem 3.1 is misleading. These type of stability estimates (of the operator mapping right hand side and initial condition to the solution) are standard tools in the theory of partial differential equations, making this seem more like a re-branding. It would strengthen the paper if the authors would discuss related results in the PDE literature.
>
> We would like to emphasize that the key message of the theorems like what we have proven here is to able to estimate the proximity of a surrogate solution from the true PDE solution in terms of errors made by the solution in satisfying the PDE in the interior of the domain considered and the errors made in satisfying the boundary and the initial conditions.
>
> To the best of our knowledge, there are scant results in conventional PDE literature which give such bounds. Kindly see the newly added Appendix E where we have reviewed various classical literature to establish that to the best of our knowledge existing results in numerical analysis cannot be deployed to understand PINN training - as is the target here. In particular, specifically for the $0$ viscosity Burgers' PDE we can only find results like this, https://doi.org/10.1090/S0025-5718-1987-0906180-5, where the theory does not give a bound on the distance from the true solution of the PDE to that found by the finite element method.
>
> More generally, in works such as Corollary 3.5 in https://doi.org/10.1137/0728048 the authors consider a weak solution of the $\varepsilon$-viscosity regularized Burgers' PDE and derive bounds on the local $L^p$-distance between the weak solution and the actual solution. There is no obvious way to apply these bounds for a PINN solution since the trained net has no guarantee to be satisfying the conditions required of the surrogate here.
>
> For results like Theorem 2.1 in https://doi.org/10.1137/0729087, we observe that these too don't have an obvious way for the bounds to be applied for PINN experiments because they need stringent conditions (like satisfying the conservativeness property) to be true for the approximant, for the bounds to apply and there is no natural way to know if the neural surrogate satisfies these conditions. Also neither of the above cited classical bounds are tailored to any compact domain and hence there is no boundary condition error that is getting tracked there as in our Theorem $3.2$.
>
>
> In contrast to such literature, even in the context of existing PINN guarantees, we posit that our theorems are a step towards novel and useful guarantees.

---

> > ### Author Response · Authors · 2023-11-15
> > **Contd.**
> >
> > >The literature review on operator learning approaches misses the works on both neural operators and BCR-NET
> >
> > Thanks for the suggestion. In the 3rd paragraph of the Introduction we have added these references.
> >
> > > Wang et al uses a re-scaled coordinate system and therefore does not actually solve a PDE with singular solution. The blow-up studied by this community is also specific to incompressible problems as the blowup of the compressible Euler equation (of which the Burgers equation is the zero sound speed limit) arises from a different phenomenon. This difference is not clear from the writing.
> >
> > We would like to emphasize that we do not claim any technical connection or dependency on the results and observations made in https://arxiv.org/abs/2201.06780. We cite that paper to make the limited point that recently there has risen evidence that PINNs might be able to get reasonable answers even in the proximity of singularity formation happening.

---

> > > ### Author Response · Authors · 2023-11-21
> > > **Thanks!**
> > >
> > > Given the rebuttal process is coming to an end in a couple of days, we would be grateful to know if the reviewer has found our answers to be satisfactorily answering all their questions. Kindly feel free to let us know if there are additional queries and we would be happy to strive to clarify them.

---

### Meta-Review · Area_Chair_15PT · 2023-12-04

**Metareview:**

This is an interesting paper that aims at analytically investigating whether PINN can characterize the finite time blow up of Burgers equation's solution. This is an important problem and the authors' effort is appreciated. However, several reviewers raised serious concerns, which persist after the rebuttal, and I agree with them. Therefore, acceptance at the current stage cannot be recommended, but I encourage the authors to consider the comments and submit a revised version in the future.

**Justification For Why Not Higher Score:**

Reviewers raised serious concerns and did not consider them resolved after the rebuttal.

**Justification For Why Not Lower Score:**

N/A

---

### Decision · Program_Chairs · 2024-01-16

Reject